# Spatial signatures of anesthesia-induced burst-suppression differ between primates and rodents

**Nikoloz Sirmpilatze[1,2,3]\*, Judith Mylius[1], Michael Ortiz-Rios[1], Jürgen Baudewig[1], Jaakko Paasonen[4], Daniel Golkowski[5,6], Andreas Ranft[7], Rüdiger Ilg[5,8], Olli Gröhn[4], Susann Boretius[1,2,3,9]\***

[1]Functional Imaging Laboratory, German Primate Center – Leibniz Institute for Primate Research, Göttingen, Germany; [2]Georg-August University of Göttingen, Göttingen, Germany; [3]International Max Planck Research School for Neurosciences, Göttingen, Germany; [4]A.I.V. Institute for Molecular Sciences, University of Eastern Finland, Kuopio, Finland; [5]Department of Neurology, Klinikum Rechts der Isar der Technischen Universität München, Munich, Germany; [6]Department of Neurology, Heidelberg University Hospital, Heidelberg, Germany; [7]Department of Anesthesiology and Intensive Care Medicine, Klinikum Rechts der Isar der Technischen Universität München, Munich, Germany; [8]Department of Neurology, Asklepios Stadtklinik Bad Tölz, Bad Tölz, Germany; [9]Leibniz Science Campus Primate Cognition, Göttingen, Germany

**\*For correspondence:**
nsirmpilatze@dpz.eu (NS);
sboretius@dpz.eu (SB)

**Competing interest:** The authors declare that no competing interests exist.

**Abstract** During deep anesthesia, the electroencephalographic (EEG) signal of the brain alternates between bursts of activity and periods of relative silence (suppressions). The origin of burst-suppression and its distribution across the brain remain matters of debate. In this work, we used functional magnetic resonance imaging (fMRI) to map the brain areas involved in anesthesia-induced burst-suppression across four mammalian species: humans, long-tailed macaques, common marmosets, and rats. At first, we determined the fMRI signatures of burst-suppression in human EEG-fMRI data. Applying this method to animal fMRI datasets, we found distinct burst-suppression signatures in all species. The burst-suppression maps revealed a marked inter-species difference: in rats, the entire neocortex engaged in burst-suppression, while in primates most sensory areas were excluded—predominantly the primary visual cortex. We anticipate that the identified species-specific fMRI signatures and whole-brain maps will guide future targeted studies investigating the cellular and molecular mechanisms of burst-suppression in unconscious states.

## Editor's evaluation

This study reveals that anesthesia-induced burst suppression's spatial patterns differ across humans, macaques, marmosets, and rats. Given that burst suppression is considered a hallmark of unconscious states, these findings are potentially important for us to understand the evolution of the neural correlates of consciousness.

## Introduction

Despite the long-standing and successful practice of general anesthesia, its underlying mechanisms remain elusive. Most anesthetics lead to a dose-dependent transition from mild sedation to deep

**eLife digest** The development of anesthesia was a significant advance in medicine. It allows individuals to undergo surgery without feeling pain or remembering the experience. But scientists still do not know how anesthesia works. During anesthesia, scientists have measured brain activity using electroencephalograms (EEG) and found that the brain appears to turn on and off. Comatose patients also have similar switches between bursts of electrical activity and periods of silence. This burst-suppression pattern may be related to unconsciousness.

But scientists still have many questions about how anesthesia causes burst-suppression. One challenge is that while an EEG can tell scientists when the brain turns on and off, it does not show exactly where this occurs. Another imaging method called functional Magnetic Resonance Imaging (fMRI) may fill this gap by allowing scientists to map where the brain activity occurs.

Sirmpilatze et al. have created detailed maps of burst-suppression in humans, primates, and rats under anesthesia by analyzing brain scans using fMRI. In rats, the entire outer layer or cortex of the brain underwent a synchronized pattern of burst-suppression. In humans and primates, areas of the brain like those responsible for eyesight did not follow the rest of the cortex in switching on and off.

The experiments reveal crucial differences in how rats and humans and other primates respond to anesthesia. The fMRI mapping technique Sirmpilatze et al. created may help scientists learn more about these differences and why some parts of human brains do not undergo burst-suppression. This may help scientists learn more about unconsciousness and help improve anesthesia or the care of comatose patients.

unconsciousness, accompanied by a progression of distinct electroencephalographic (EEG) patterns (*Brown et al., 2011*; *Brown et al., 2010*). This progression is well characterized for halogenated volatile agents, like isoflurane (*Kroeger et al., 2013*). Low isoflurane concentrations lead to high-amplitude low-frequency activity, whereas at higher concentrations, the ongoing activity is quasi-periodically interrupted by quiescent periods that can last from seconds to minutes. This alternation between bursts of activity and gaps of quiescence is termed burst-suppression (*Swank, 1949*; *Swank and Watson, 1949*). As anesthesia deepens, suppression periods become longer and ultimately culminate in complete electrical silence. The same sequence unfolds for many other anesthetics (*Akrawi et al., 1996*; *Fleischmann et al., 2018*) and even during deep hypothermia (*Westover et al., 2015*; *Zhang et al., 2010*). Burst-suppression also appears in comas of various etiologies, for which it holds diagnostic and prognostic value (*Brenner, 1985*; *Brown et al., 2010*; *Cloostermans et al., 2012*; *Hofmeijer et al., 2014*; *Young, 2000*). Clinically, burst-suppression is often used as the target EEG pattern for intended neuroprotection as well as pharmacological control of intracranial hypertension (*An et al., 2015*; *Westover et al., 2015*).

The presence of burst-suppression in a variety of unconscious conditions has led some researchers to propose that it represents a common low-order 'attractor' state—a hallmark of a profoundly inactive brain (*Ching et al., 2012*). Several models have focused on the temporal bistability (*Bojak et al., 2015*; *Ching et al., 2012*; *Kroeger and Amzica, 2007*; *Liley and Walsh, 2013*), while many open questions surround the spatial properties of bursts—namely, their origin and propagation across brain areas. Bursts have traditionally been viewed as globally synchronous events, based on their simultaneous occurrence across EEG electrodes (*Clark and Rosner, 1973*; *Swank, 1949*). However, recent findings from electrocorticogram in humans (*Lewis et al., 2013*) and widefield calcium imaging in rats (*Ming et al., 2020*) have challenged this view by demonstrating considerable spatial variation in burst origin and propagation. Furthermore, it is yet unclear whether burst-suppression is intrinsically cortical or is influenced by subcortical sources. According to one hypothesis, the cortex spontaneously falls into burst-suppression upon anatomical or functional disconnection from other brain areas. This view is supported by observations of burst-suppression in isolated neocortical slice preparations (*Lukatch et al., 2005*; *Lukatch and MacIver, 1996*), but contradicted by the ability of sensory stimuli to evoke bursts indistinguishable from the spontaneously occurring ones (*Hartikainen et al., 1995*; *Hudetz and Imas, 2007*; *Kroeger and Amzica, 2007*; *Land et al., 2012*).

Incorporating detailed spatial information into burst-suppression models could reduce the pool of possible mechanisms. Such information, in the form of three-dimensional whole-brain maps, can be

provided by blood-oxygen-level-dependent (BOLD) functional magnetic resonance imaging (fMRI, *Logothetis and Pfeuffer, 2004*; *Ogawa et al., 1990*). The two distinct phases of burst-suppression correspond to different energy consumption levels, which should—through neurovascular coupling—translate into noticeable changes in blood flow and oxygenation (*Hillman, 2014*). This premise has been confirmed in several species. In rats, a combination of EEG with optical imaging of cerebral blood flow revealed that bursts are coupled with strong hemodynamic responses (*Liu et al., 2011*). The same effect was demonstrated in macaques (*Zhang et al., 2019*) and humans (*Golkowski et al., 2017*) using simultaneous EEG-fMRI. These two studies have shown that the burst timing from EEG can be used for fMRI mapping of burst-suppression.

While the preservation of neurovascular coupling during burst-suppression holds promise for clinical research, it creates pitfalls for the increasingly popular animal fMRI studies. About four of five such studies are conducted under anesthesia, most commonly with isoflurane (*Mandino et al., 2020*). Given that medium-to-high concentrations of isoflurane lead to burst-suppression (*Kroeger et al., 2013*), we hypothesized that many animal fMRI experiments have been conducted during this state. This assumption is supported by reports of global BOLD signal synchronization in isoflurane-anesthetized animals (*Kalthoff et al., 2013*; *Liu et al., 2013*; *Paasonen et al., 2018*), as is expected during burst-suppression (*Zhang et al., 2019*). Animal fMRI researchers typically try to avoid this confounding effect by using low concentrations of isoflurane (*Liu et al., 2013*), opting for different anesthetic regimes (*Kalthoff et al., 2013*; *Paasonen et al., 2018*), or applying global signal regression during fMRI preprocessing (*Kalthoff et al., 2013*; *Liu et al., 2013*; *Zhang et al., 2019*).

Importantly, burst-suppression is not a nuisance variable, but a biologically and clinically significant neural phenomenon. Here, we show that fMRI data from anesthetized humans and animals can be exploited to map the spatial distribution of burst-suppression using a data-driven approach. Unlike previous studies, our strategy applies the same methodology across species, allowing us to examine whether specific brain regions are involved universally across mammals. To achieve this, we first implemented a simple heuristic algorithm for identifying the fMRI signatures of burst-suppression in a human EEG-fMRI dataset (*Golkowski et al., 2017*; *Ranft et al., 2016*). We then used this algorithm to detect instances of burst-suppression in fMRI data from anesthetized animals (macaques, marmosets, and rats) and constructed species-specific whole-brain maps of burst-suppression. These maps provide a detailed spatial account of burst-suppression in some of the most widely used animal models and reveal crucial differences between primates and rodents. The maps will serve as a guide for identifying promising target regions and facilitate translational research into the mechanisms of burst-suppression and its role in brain pathophysiology.

## Results

### fMRI signatures of EEG-defined burst-suppression in anesthetized humans

Simultaneous EEG-fMRI recordings are ideal for constructing a whole-brain map of burst-suppression (*Golkowski et al., 2017*; *Zhang et al., 2019*). EEG supplies the timing of burst and suppression phases, which is convolved with a hemodynamic response function (HRF) and correlated with fMRI to produce voxel-wise statistical parametric maps. However, simultaneous EEG-fMRI recordings are challenging, particularly at high magnetic field strengths. Knowing the exact hemodynamic correlates of burst-suppression may circumvent the need for concurrent EEG, especially if these correlates are specific enough to serve as a unique fMRI signature.

To explore this approach, we revisited an EEG-fMRI dataset acquired in anesthetized human participants (*Ranft et al., 2016*), which contains task-free measurements at three different concentrations of the anesthetic sevoflurane: 3.9–4.6% ('high'), 3% ('intermediate'), and 2% ('low'). Importantly, this concentration range spans across EEG states, allowing us to examine how they differ in their fMRI patterns. We categorized the EEG recordings as one of two states—burst-suppression or continuous slow-wave activity (uninterrupted by suppressions). The high sevoflurane concentration had been intentionally chosen in the initial study to induce burst-suppression (*Ranft et al., 2016*), and indeed 19/20 subjects exhibited the typical bistable EEG alternation at this concentration (*Figure 1A*). The same applied to 4/20 subjects at the intermediate concentration, albeit with shorter suppression phases. The remaining intermediate-concentration recordings as well as all low-concentration

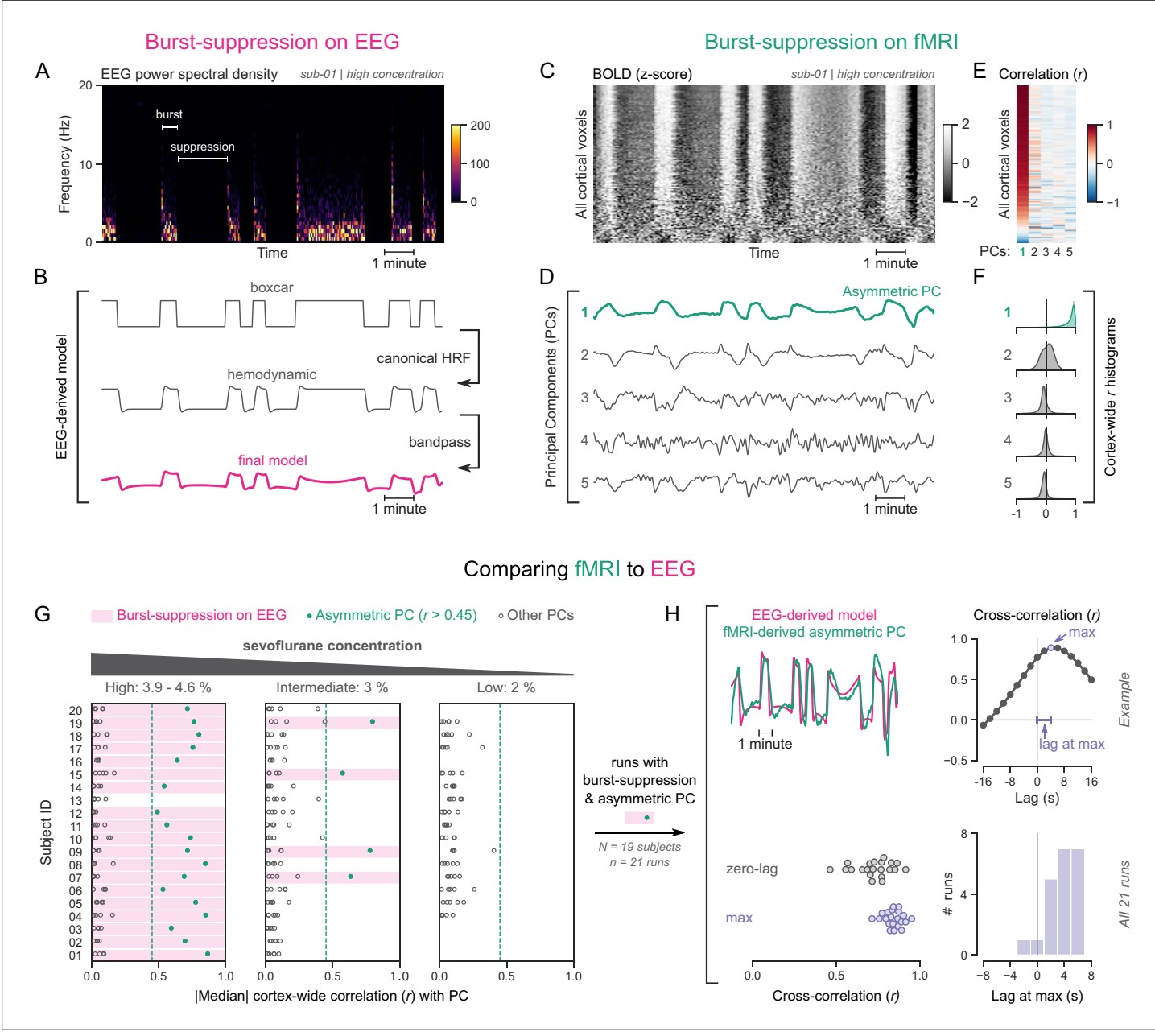

**Figure 1.** Functional magnetic resonance imaging (fMRI) signatures of electroencephalogram (EEG)-defined burst-suppression in anesthetized humans. (**A**) Burst and suppression phases are marked on an EEG spectrogram of a human participant under sevoflurane anesthesia. (**B**) To derive the hemodynamic model of burst-suppression, the above phases (boxcar function) are convolved with a hemodynamic response function (HRF) and bandpass filtered (0.005–0.12 Hz). (**C**) The cortical blood-oxygen-level-dependent (BOLD) signal from the concurrent fMRI is represented as a two-dimensional matrix (carpet plot). The rows (voxels) are ordered according to their correlation with the mean cortical signal. (**D**) The first five temporal principal components (PCs) of the shown matrix are plotted. Cortex-wide Pearson's correlation coefficients (r) are shown for the first PCs, both as a heatmap (**E**) and as histograms for each PC. (**F**) Taken together, panels C–F demonstrate that burst-suppression manifests as a widespread cortical BOLD signal fluctuation captured by the first PC. This component (PC1), unlike the rest, is positively correlated with most cortical voxels and is referred to as an 'asymmetric PC'. *Figure 1—figure supplement 1* shows a counterexample, that is an fMRI run acquired during the continuous slow-wave state that exhibits no asymmetric PCs. (**G**) To demonstrate the correspondence between burst-suppression on EEG and asymmetric PCs on fMRI across the entire human dataset, median cortex-wide r values of the first five PCs are plotted as dots for each run. The runs that exhibit burst-suppression on EEG also have a single prominent asymmetric PC with median cortex-wide r>0.45 (except for subject 15 at high concentration). (**H**) For these runs, the cross-correlation between EEG-derived hemodynamic models and asymmetric PCs is shown. The cross-correlation at zero-lag, the maximum cross-correlation, and the time-lag at the maximum are extracted for each run (see example in top row) and plotted across all runs (bottom row). Numerical data for panels G and H are provided in *Figure 1—source data 1*.

*Figure 1 continued on next page*

Figure 1 continued

The online version of this article includes the following source data and figure supplement(s) for figure 1:

**Source data 1.** Numerical data for *Figure 1G–H*.

**Figure supplement 1.** A functional magnetic resonance imaging (fMRI) run without asymmetric principal components (PCs).

recordings showed uninterrupted slow-wave EEG activity. These results are consistent with the known dose-dependency of burst-suppression (*Pilge et al., 2014*).

To examine the potential effects of the two aforementioned EEG states on fMRI time series, we used the so-called 'grayplot' or 'carpet plot'—a two-dimensional (2D) representation of fMRI data suited for detecting global signal fluctuations (*Aquino et al., 2020*; *Power, 2017*). The carpet plot is a heatmap of BOLD signal intensity, where each row represents a normalized (z-score) voxel time series. Upon examining carpet plots from fMRI runs with EEG-defined burst-suppression, we identified pronounced signal fluctuations that spanned almost the entire cortex (*Figure 1C*). Conversely, fMRI runs with continuous slow-wave EEG activity showed no such cortex-wide fluctuations (see *Figure 1— figure supplement 1*).

Next, we sought to quantify the exact correspondence between widespread BOLD fluctuations and EEG burst-suppression. We performed principal component analysis (PCA) on the 2D matrix of cortical voxel time series, extracted the first five temporal components, and correlated them with the voxel time series. In fMRI runs with cortex-wide fluctuations, the bistable pattern visible on the carpet plot was captured by one of the principal components (PCs)—in most cases the first PC (*Figure 1D*). This component was strongly correlated with most cortical voxel time series (*Figure 1E*), shifting the histogram of Pearson's correlation coefficients ($r$) away from zero—unlike the symmetric zero-centered histograms of other PCs (*Figure 1F*). Due to this property, we will hereafter refer to such components as 'asymmetric' PCs and use the median of the cortex-wide $r$ values as a measure of asymmetry. The presence of asymmetric PCs proved to be a reliable signature of burst-suppression, as we demonstrate in *Figure 1G* across subjects. The two EEG-defined states (burst-suppression vs continuous slow waves) were linearly separable based on the median $r$ value alone. Every run with EEG burst-suppression had a single PC with median $r>0.45$, which was not the case for any of the runs with continuous slow waves. A single recording was an exception to this rule (subject 15 at high sevoflurane concentration, see *Figure 1G*), exhibiting no asymmetric PC. The EEG of this recording showed near-constant suppression interrupted by only one short burst.

So far, we have seen that burst-suppression is uniquely associated with an asymmetric PC that captures a widespread fluctuation in the cortical BOLD signal. Notably, this asymmetric PC closely follows the alternating burst-suppression pattern of EEG (*Figure 1A–D*) and likely is a direct hemo-dynamic correlate. To verify this last observation, we compared each asymmetric PC with a neurovas-cular model derived from the concurrent EEG recording. For constructing the model, we convolved the EEG pattern (1 for burst, 0 for suppression) with a canonical HRF (*Glover, 1999*) and passed the resulting time series through the bandpass filter used for fMRI preprocessing (0.005–0.12 Hz, see *Figure 1B*). The BOLD signal of the asymmetric PCs was indeed strongly correlated with the EEG-derived model (Pearson's $r=0.73 \pm 0.11$, mean ± SD), but with a notable time-lag between the two. A cross-correlation analysis (*Figure 1H*) showed that the asymmetric PC was delayed by $3.71 \pm 2.22$ s (mean ± SD) compared with the neurovascular model. This lag can be attributed to a slowing of the HRF—an expected effect of anesthesia (*Gao et al., 2017*). We accounted for this lag by comparing the asymmetric PCs to the EEG-derived models at the time lags that maximized their cross-correlation and found that the correlation increased to $r=0.84 \pm 0.06$.

## Mapping burst-suppression in anesthetized humans without EEG

Since asymmetric PCs are the hemodynamic correlates of burst-suppression, their spatial distribution should represent a whole-brain map of burst-suppression. The map should be equivalent to the one obtained via the existing approach—i.e., using the EEG-derived models as regressors in a general linear model (GLM, *Golkowski et al., 2017*). We verified this by performing two GLM analyses for each fMRI run with burst-suppression—one with the asymmetric PC and one with the EEG-derived model (without time lag correction). As expected, the spatial patterns of the resulting Z statistical

maps were nearly identical between the two analyses: neighborhood cross-correlation $r=0.96 \pm 0.03$ (*Figure 2A*).

To map the spatial distribution of burst-suppression across subjects, we performed a second-level group GLM using the asymmetric PCs as regressors. The group statistics were carried out in the MNI152 template space (*Grabner et al., 2006*) with the FSL (FMRIB's Software Library) tool randomise (*Winkler et al., 2014*). The resulting *T* statistic maps were thresholded using threshold-free cluster enhancement (TFCE, *Smith and Nichols, 2009*) and a corrected p<0.05 (*Figure 2B*). The group map revealed significant correlation with asymmetric PCs (burst-suppression) in the striatum and across most of the cortex, with the prominent exception of occipital areas in and around the calcarine sulcus—the location of primary visual cortex (V1). In addition to V1, several other cortical patches were non-correlated with burst-suppression. We visualized these on an inflated representation of the cortical surface (*Figure 2C–D*). Some of them overlapped with primary cortices: somatosensory and motor areas around the central sulcus, and auditory areas on the dorsal bank of superior temporal gyrus—including Heschl's gyrus. Additional inactive patches included the subcallosal cortex (subgenual cingulate), the parahippocampal gyrus, and the border between the insula and the frontoparietal operculum. *Figure 2—figure supplement 1* provides a closer look at subcortical structures. Anterior and midline parts of the thalamus were significantly correlated with burst-suppression, whereas the posterior thalamic nuclei, the cerebellar cortex, the hippocampus, and the amygdala were, for the most part, not engaged.

Intriguingly, we observed significant anticorrelation with burst-suppression at the ventricular borders. After a closer examination of this effect (see *Figure 2—figure supplement 2*), we believe it relates to a cycle of ventricular shrinkage (during bursts) and expansion (during suppressions). The widespread hemodynamic fluctuations during burst-suppression may exert a 'pumping' effect on the ventricles—an explanation consistent with reports of cerebrospinal fluid (CSF) oscillations during sleep (*Fultz et al., 2019*).

## Nonhuman primates exhibit human homologous fMRI signatures of burst-suppression

Having identified an fMRI signature of burst-suppression in the human data, we searched for similar signatures in nonhuman primate fMRI data. We first evaluated an fMRI dataset of female long-tailed macaques (*Macaca fascicularis*), acquired at 3T. Macaques are phylogenetically close to humans, have a long tradition of being used for neuroscientific research, and account for most fMRI studies in nonhuman primates (*Mandino et al., 2020*; *Milham et al., 2020*). The 13 macaques included in our study were anesthetized with isoflurane (concentration range 0.95–1.5%) and mechanically ventilated. For each macaque, one or two fMRI runs (each 10–20 min in duration) were acquired within a single imaging session.

We analyzed each fMRI run using the same approach as for human fMRI data—carpet plots and PCA. Several fMRI runs clearly exhibited the burst-suppression signature: a widespread bistable fluctuation on the carpet plot that was always captured by an asymmetric PC (see example in *Figure 3A*, counterexample in *Figure 3—figure supplement 1*). *Figure 3B* shows the cortex-wide median *r* value for the first five PCs across all fMRI runs. We selected nine runs (from 7/13 subjects) with a single prominent asymmetric PC (median *r*>0.15) and used these PCs as regressors for GLM. The group statistical map (*Figure 3C–E*) exhibited striking homology with the human burst-suppression map. Significantly correlated areas included the striatum and most of the cortex, with the following exceptions: V1, extrastriate visual areas, parts of somatosensory and motor cortices on either side of the central sulcus, the subcallosal cortex, and the parahippocampal gyrus. One notable difference to the human map was the auditory cortex, the entirety of which was among the positively correlated areas. The cerebellar cortex was not correlated with burst-suppression, while the thalamus, hippocampus, and amygdala were partly involved (see *Figure 3—figure supplement 2*). No significant anticorrelations were found around the ventricles or elsewhere.

We next aimed to reproduce our findings in the common marmoset (*Callithrix jacchus*). The layout of areas in the marmoset brain closely resembles that of other primates (*Liu et al., 2018*), including macaques and humans. Yet, the marmosets' body size is comparable to rats, and they fit into high-field small animal MRI systems that are routinely used for imaging rodents. These properties make the marmoset an ideal model for bridging neuroimaging results between larger primates and rodents.

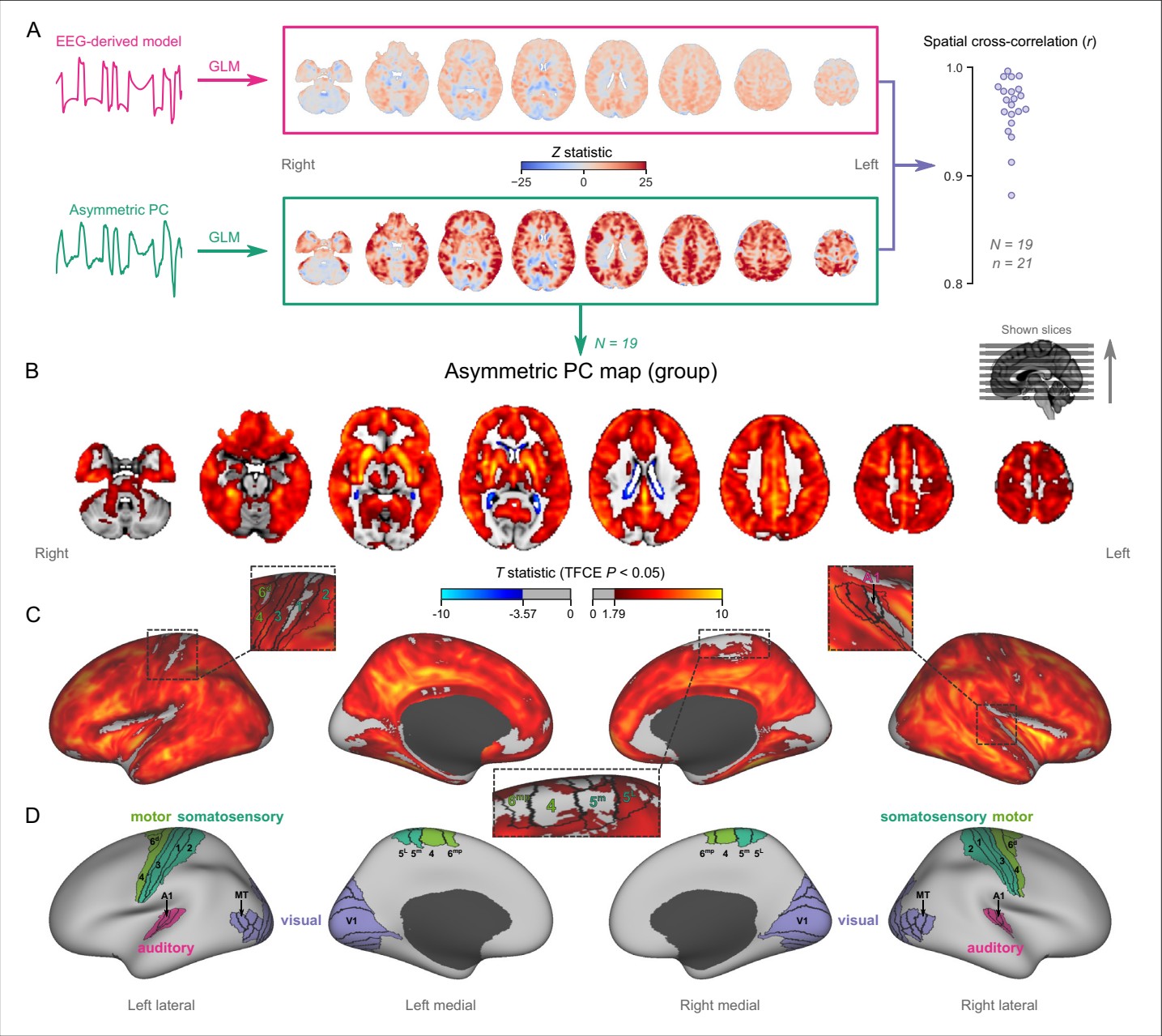

**Figure 2.** Mapping burst-suppression in anesthetized humans without electroencephalogram (EEG). (**A**) Maps of burst-suppression are computed via general linear model (GLM) analysis using one of two regressors—either the EEG-derived hemodynamic model or the functional magnetic resonance imaging (fMRI)-derived asymmetric principal component (PC). The resulting Z statistic maps, for an example subject are shown here in the MNI152 template space. Neighborhood cross-correlation values between the two types of Z statistic maps are plotted on the right, across all runs with asymmetric PCs (N=19 subjects, n=21 EEG-fMRI runs, see *Figure 2—source data 1*). (**B**) The group burst-suppression map, computed via a second-level analysis of the single-subject asymmetric PC GLMs, is shown here overlaid on the MNI152 volumetric template. The group statistics were carried out with the FSL (FMRIB's Software Library) tool randomise; the resulting T statistic maps were thresholded using threshold-free cluster enhancement (TFCE) and a corrected p<0.05. *Figure 2—figure supplement 1* provides a closer look at subcortical structures, while *Figure 2—figure supplement 2* examines the source of the observed anticorrelation at the ventricular borders. (**C**) The group burst-suppression map is shown in fsaverage surface space. Non-cortical areas on the medial surface are shown in dark gray. (**D**) The locations of several sensory and motor cortical areas, based on the Human Connectome Project multimodal parcellation, are indicated on the surface: primary motor (area 4), premotor (areas 6d and 6mp), primary somatosensory (areas 3a–b, 1, and 2), higher somatosensory (areas 5 m and 5 L), primary auditory, higher auditory (medial and lateral belt, parabelt), primary visual, and higher-order visual (V2, V3, V3A, V3B, V4, V4t, V6, V6A, V7, V8, MT, MST, and lateral occipital areas 1–3). *Figure 2—figure supplement 3* shows the unthresholded group burst-suppression map in both volumetric and surface representations. *Figure 2—figure supplement 4* shows a temporal signal-to-noise ratio map overlaid on the volumetric template.

*Figure 2 continued on next page*

*Figure 2 continued*

The online version of this article includes the following source data and figure supplement(s) for figure 2:

**Source data 1.** Spatial cross-correlation values for *Figure 2A*.

**Figure supplement 1.** A closer look at subcortical structures.

**Figure supplement 2.** Ventricular motion during burst-suppression.

**Figure supplement 3.** Unthresholded asymmetric principal component (PC) map.

**Figure supplement 4.** Temporal signal-to-noise ratio (tSNR) map.

For this study, we used fMRI data from 20 common marmosets (10/20 females), imaged at 9.4 T. The animals were anesthetized with isoflurane (concentration range 0.6–1.1%) and mechanically ventilated. One fMRI run (10 min in duration) was acquired per animal.

Analyzing the data as described for humans and macaques, we discovered a subset of marmoset fMRI runs that exhibited the burst-suppression signature (see example in *Figure 4A*, counterexample in *Figure 4—figure supplement 1*). We selected 8/20 subjects with a single prominent asymmetric PC (median cortex-wide $r>0.22$) and used these PCs as GLM regressors (*Figure 4B*). The group statistical map was overlaid on the National Institutes of Health (NIH) population template (*Liu et al., 2021*). The resulting map (*Figure 4C–E*) was functionally homologous to the macaque map. Positively correlated areas comprised the striatum and a large part of the cortex—including auditory areas, but excluding V1, extrastriate visual areas, parts of somatosensory and motor cortices, and most of the parahippocampal gyrus. The cerebellar cortex and the thalamus (except for a small anterior part) were not correlated with burst-suppression, while the hippocampus and the amygdala were correlated with it in parts (see *Figure 4—figure supplement 2*). No significant anticorrelations were found.

The fMRI field of view in marmosets left out a large posterior portion of V1 (see *Figure 4*). To ensure that the entirety of V1 was uncorrelated with asymmetric PCs, we obtained fMRI data with whole-brain coverage from one additional marmoset (four fMRI runs, each lasting 5 min, acquired at an isoflurane concentration of 1.1%). All four fMRI runs exhibited asymmetric PCs, which were used to compute a map via a fixed-effects GLM analysis. The map showed lack of correlation across V1, including the posterior parts that were previously excluded from the field of view (see *Figure 4—figure supplement 5*).

## Rats exhibit pancortical fMRI signatures of burst-suppression

Finally, we asked how our observations translate to rodents, specifically rats—the most popular animal model in preclinical fMRI (*Mandino et al., 2020*). Several studies have identified burst-suppression in electrophysiological recordings of rats anesthetized with isoflurane, mostly at concentrations between 1.3 and 2% (*Detsch et al., 2002*; *Hudetz and Imas, 2007*; *Liu et al., 2011*; *Masamoto et al., 2009*; *Stenroos et al., 2021*). One study described the potential fMRI correlate of burst-suppression in rats: widespread BOLD synchrony across the cortex and the striatum (*Liu et al., 2011*). Therefore, we anticipated at which isoflurane concentrations burst-suppression might occur, and how the maps might look like. To replicate these findings, we obtained rat fMRI data covering the concentration range between 1.5 and 2.5% isoflurane. The animals (11 adult female Wistar rats) were mechanically ventilated and imaged at 9.4T. We first acquired three fMRI runs at 2% isoflurane—each covering 12 min, and then switched to either a lower (1.5%, in 6/11 rats) or a higher (2.5%, in 5/11 rats) concentration before acquiring three additional fMRI runs.

We followed the same analysis procedure as in primates. *Figure 5A* shows an example fMRI run with an asymmetric PC (a counterexample can be found in *Figure 5—figure supplement 1*). We found multiple asymmetric PCs across all fMRI runs (*Figure 5B*). To select the ones most resembling burst-suppression, we focused on the subset of runs containing at least one asymmetric PC with median cortex-wide $r>0.2$ (37/64) but excluded nine runs that also had a second PC with relatively high median $r$ value (within $r=0.15$ of the most asymmetric PC). We were left with 28 runs, 24 of which were acquired at 2% isoflurane and 4 at 1.5%. We used the 28 asymmetric PCs as regressors in a GLM and constructed the group-level statistical map. The map (*Figure 5C–E*) showed significant correlation with the asymmetric PCs in the striatum and the entire neocortex. Contrary to primates, the map included all primary sensory and motor cortices. The cerebellar cortex and the amygdala did not correlate with burst-suppression, while significant correlations were found in parts of the thalamus

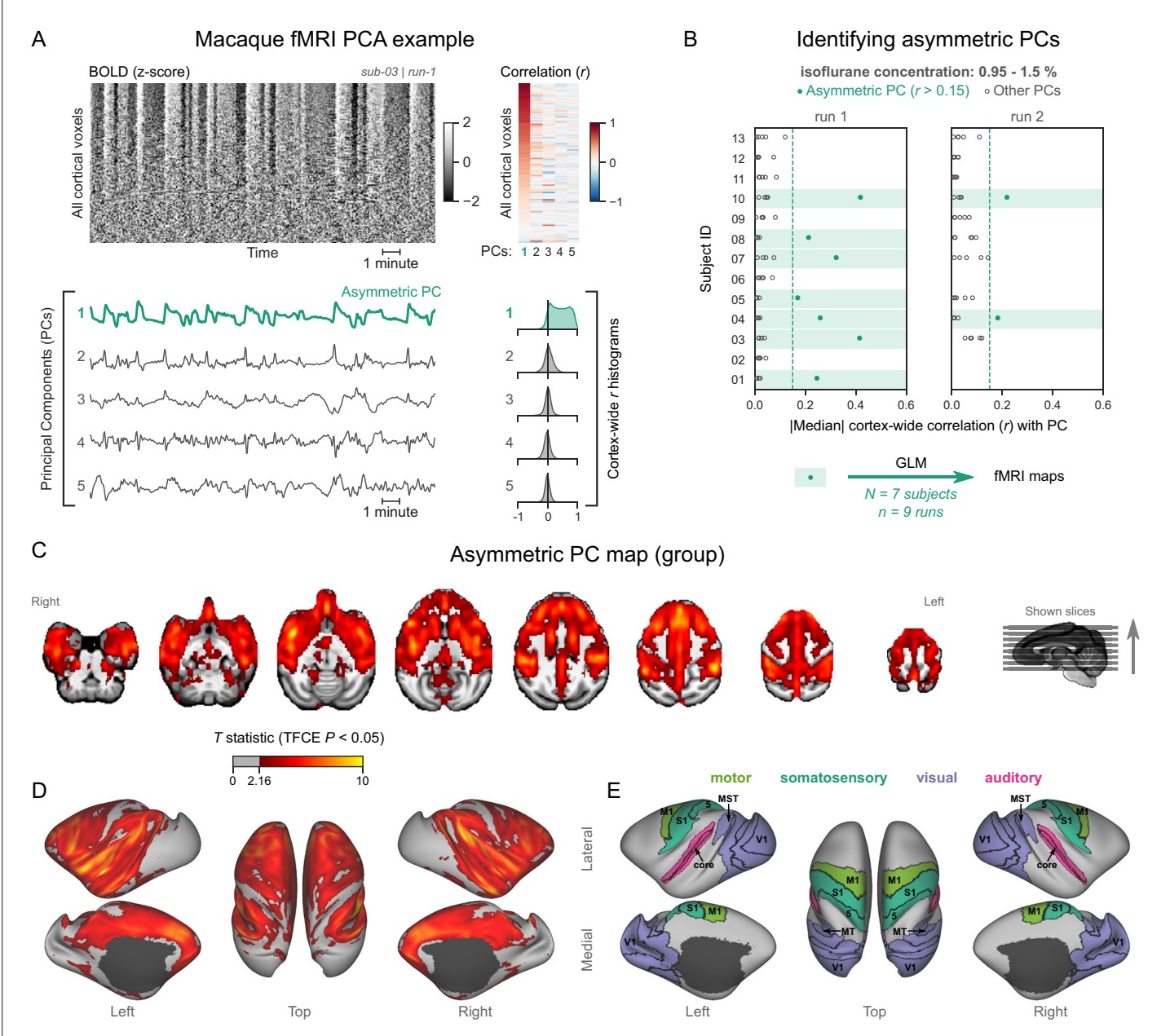

**Figure 3.** Macaques exhibit human homologous functional magnetic resonance imaging (fMRI) signatures of burst-suppression. (**A**) The cortical blood-oxygen-level-dependent (BOLD) fMRI signal of a long-tailed macaque during isoflurane anesthesia is represented as a carpet plot. The rows (voxels) are ordered according to their correlation with the mean cortical signal. The first five temporal principal components (PCs) of the signal are plotted below. The Pearson's correlation coefficients (*r*) between the PCs and all cortical voxels are represented both as a heatmap and as histograms for each PC. The first PC captures the widespread fluctuation visible on the carpet plot and has an asymmetric *r* histogram. *Figure 3—figure supplement 1* shows a counterexample, that is an fMRI run that exhibits no asymmetric PCs. (**B**) Cortex-wide median *r* values for the first five PCs are plotted as dots across the entire macaque dataset (see *Figure 3—source data 1*). The fMRI runs with a single prominent asymmetric PC (*r*>0.15, highlighted in green) are selected for general linear model (GLM) analysis. (**C**) The group asymmetric PC map, computed via a second-level analysis of single-subject GLMs, is shown here overlaid on a study-specific volumetric template. The group statistics were carried out with FSL randomise; the resulting *T* statistic maps were thresholded using threshold-free cluster enhancement (TFCE) and a corrected p<0.05. *Figure 3—figure supplement 2* provides a closer look at subcortical structures. (**D**) The same group map is shown on a cortical surface representation of the template. Non-cortical areas on the medial surface are shown in dark gray. (**E**) The locations of several sensory and motor cortical areas, based on the cortical hierarchical atlas of the rhesus macaque, are indicated on the surface: primary visual (V1), higher-order visual (V2, V3, V4, V6, MT, MST, and FST), primary somatosensory (S1), higher somatosensory (area 5), primary motor (M1), and auditory cortices (auditory core, belt, and parabelt). *Figure 3—figure supplement 3* shows the unthresholded group

*Figure 3 continued on next page*

*Figure 3 continued*

asymmetric PC map in both volumetric and surface representations. *Figure 3—figure supplement 4* shows a temporal signal-to-noise ratio map overlaid on the volumetric template.

The online version of this article includes the following source data and figure supplement(s) for figure 3:

**Source data 1.** Numerical data for *Figure 3B*.

**Figure supplement 1.** A functional magnetic resonance imaging (fMRI) run without asymmetric principal components (PCs).

**Figure supplement 2.** A closer look at subcortical structures.

**Figure supplement 3.** Unthresholded asymmetric principal component (PC) map.

**Figure supplement 4.** Temporal signal-to-noise ratio (tSNR) map.

**Figure supplement 4—source data 1.** Numerical data for *Figure 3—figure supplement 4B*.

(medial) and the hippocampus (see *Figure 5—figure supplement 2*). No significant anticorrelations were found.

The burst-suppression map shown in *Figure 5* resembles a unified cortico-striatal network often found by BOLD fMRI studies in isoflurane-anesthetized rats (*Kalthoff et al., 2013*; *Liu et al., 2013*; *Liu et al., 2011*; *Paasonen et al., 2018*; *Williams et al., 2010*), suggesting that previous studies were potentially conducted during burst-suppression. To reinforce this notion, we revisited the data (six rats, anesthetized with 1.3% isoflurane) from one study (*Paasonen et al., 2018*) and subjected it to our preprocessing and analysis pipeline. As shown in *Figure 5—figure supplement 5*, we found a single asymmetric PC (median cortex-wide $r$>0.2) in all rats, with spatial distribution that matched both the unified network described in the original study (*Paasonen et al., 2018*) and the rat burst-suppression map in the present manuscript. The results strongly suggest that the apparent unification of cortex and striatum into a single functional network is a direct effect of burst-suppression.

## Primate V1 is uncoupled from the rest of the cortex during burst-suppression

Our results so far revealed a striking difference between primates and rodents: burst-suppression in rats engaged the entire neocortex, while in the three primate species, it appeared to largely spare sensory cortices. The most prominent and unambiguous example is V1, which in primates showed no correlation to the asymmetric PCs. This finding cannot be attributed to a fluke of thresholding (see unthresholded *T* statistic maps in figure supplement 3 of *Figures 2–5*) or to regional differences in temporal signal-to-noise ratio (tSNR, see figure supplement 4 of *Figures 2–5*). The visual cortex of the macaque brain did exhibit lower tSNR compared with more anterior areas, but this fact alone cannot account for its sparing. Other macaque brain areas with comparably low tSNR values, such as area TE of the inferior temporal cortex, were significantly correlated with asymmetric PCs (*Figure 3—figure supplement 4*).

The sparing of V1 prompted us to look more closely at its activity and to compare it with an area strongly engaged in burst-suppression. In each of the four species, we defined two regions-of-interest—one in V1 and another in the cingulate cortex (Brodmann area 24 in primates, primary cingulate cortex in rats)—and extracted their mean BOLD signal time series from fMRI runs, previously identified as burst-suppression. We then computed the SD of the time series, as a measure of BOLD signal amplitude, and its power spectral density (PSD). We also performed seed-based correlation analysis using the extracted time series as GLM regressors. The results are shown in *Figure 6* across species. As expected, the cingulate exhibited the same bistable fluctuation as the asymmetric PCs, and its seed-based correlation map was identical to the species-specific asymmetric PC maps. V1 activity in rats was indistinguishable from that of the cingulate across all metrics: BOLD signal amplitude (p=0.63, paired samples two-sided Wilcoxon rank-sum test), PSD, and seed-based correlation map. Conversely, V1 activity in primates was very different from the cingulate: BOLD signal amplitude was reduced (p<0.001 in humans, p=0.004 in macaques, p=0.008 in marmosets; paired samples two-sided Wilcoxon rank-sum test), PSD was lower across all frequencies, and seed-based correlation maps were confined to the V1 itself, barely extending beyond the seed borders.

We can conceive three possible mechanisms for the apparent uncoupling of V1 in primates: V1 is engaged in continuous slow waves without suppressions, it is in constant suppression, or there is

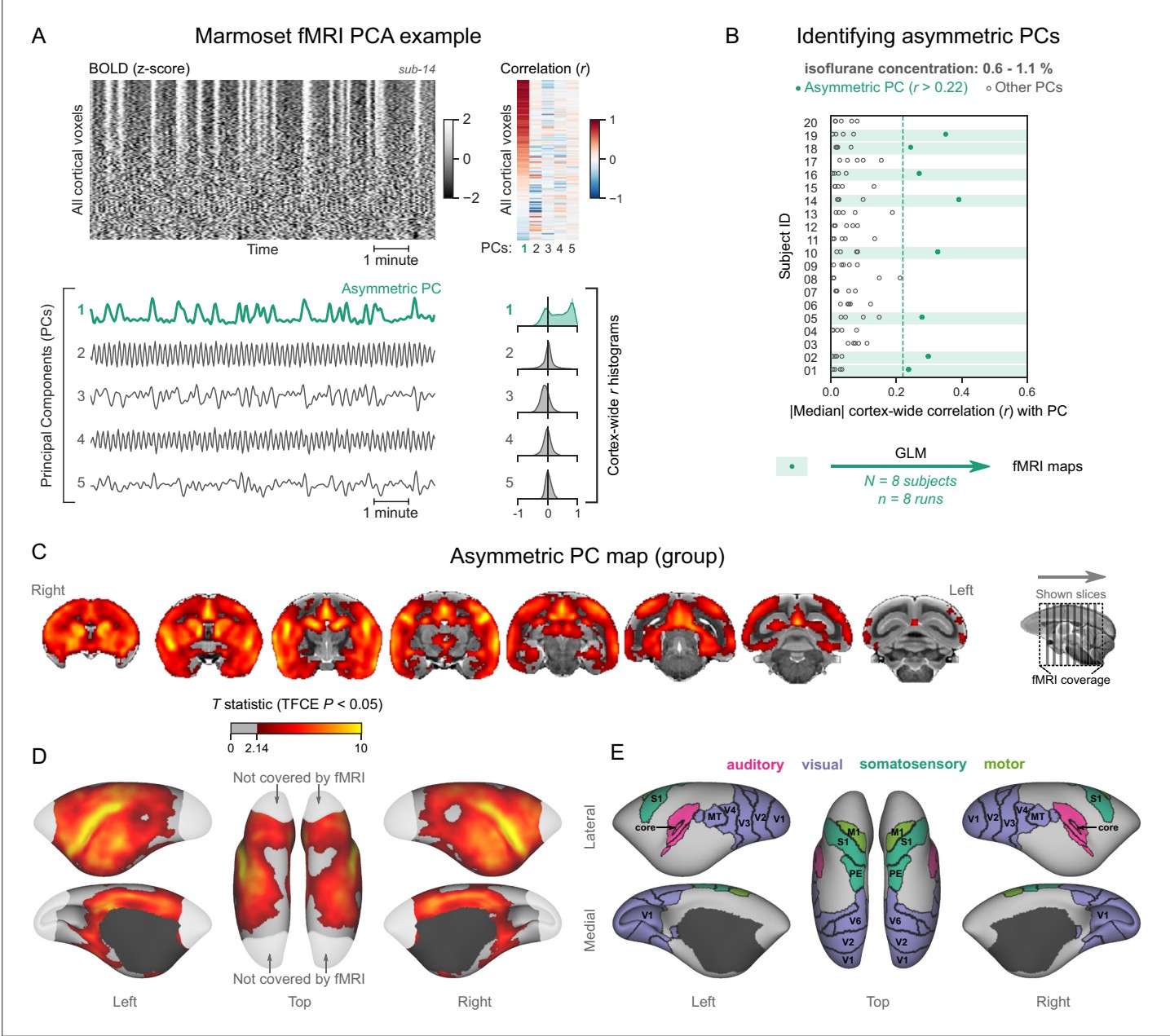

**Figure 4.** Marmosets exhibit human homologous functional magnetic resonance imaging (fMRI) signatures of burst-suppression. (**A**) The cortical blood-oxygen-level-dependent (BOLD) fMRI signal of a common marmoset during isoflurane anesthesia is represented as a carpet plot, with rows (voxels) ordered according to their correlation with the mean cortical signal. The first five temporal principal components (PCs) of the signal are plotted below. The Pearson's correlation coefficients (r) between the PCs and all cortical voxels are represented both as a heatmap and as histograms for each PC. The first PC captures the widespread fluctuation visible on the carpet plot and has an asymmetric r histogram. *Figure 4—figure supplement 1* shows a counterexample, an fMRI run that exhibits no asymmetric PCs. (**B**) Cortex-wide median r values for the first five PCs are plotted as dots across the entire marmoset dataset (see *Figure 4—source data 1*). The fMRI runs with a single prominent asymmetric PC (r>0.22, highlighted in green) are selected for general linear model (GLM) analysis. (**C**) The group asymmetric PC map, computed via a second-level analysis of single-subject GLMs, is shown here overlaid on the NIH v3.0 population template (T2-weighted image). The group statistics were carried out with FSL randomise; the resulting T statistic maps were thresholded using threshold-free cluster enhancement (TFCE) and a corrected p<0.05. *Figure 4—figure supplement 2* provides a closer look at subcortical structures. (**D**) The same group map is shown on a cortical surface representation of the template. Non-cortical areas on the medial surface are shown in dark gray. Areas not covered by the fMRI volume are shown in white. (**E**) The locations of several sensory and motor cortical areas, according to the NIH MRI-based cortical parcellation, are indicated on the surface: primary visual (V1), higher-order visual (V2, V3, V4, V6, MT, MST, and Brodmann area 19), primary somatosensory (S1), higher somatosensory (area PE), primary motor (M1), and auditory cortices (auditory core, belt, parabelt, and superior temporal rostral area). *Figure 4—figure supplement 3* shows the unthresholded group asymmetric PC map in both volumetric

*Figure 4 continued on next page*

*Figure 4 continued*

and surface representations. *Figure 4—figure supplement 4* shows a temporal signal-to-noise ratio map overlaid on the volumetric template. *Figure 4—figure supplement 5* shows results from an additional marmoset in which the whole brain was covered by the fMRI volume.

The online version of this article includes the following source data and figure supplement(s) for figure 4:

**Source data 1.** Numerical data for *Figure 4B*.

**Figure supplement 1.** A functional magnetic resonance imaging (fMRI) run without asymmetric principal components (PCs).

**Figure supplement 2.** A closer look at subcortical structures.

**Figure supplement 3.** Unthresholded asymmetric principal component (PC) map.

**Figure supplement 4.** Temporal signal-to-noise ratio (tSNR) map.

**Figure supplement 5.** Marmoset burst-suppression map with whole-brain coverage.

**Figure supplement 5—source data 1.** Numerical data for *Figure 4—figure supplement 5B*.

a local change in neurovascular coupling. In the previous analysis of the human dataset (*Golkowski et al., 2017*), the occipital electrodes of the EEG showed the same bursts as the more anterior electrodes, which might imply a true uncoupling between the electrical activity and the BOLD signal in V1. However, EEG may simply lack the spatial resolution to record a different electrical activity originating in V1: the small V1 area on the lateral cortical surface is dwarfed by non-visual areas engaging in burst-suppression (*Figure 2*). Lacking invasive neural recordings, we cannot dissect the above possibilities with certainty.

Nevertheless, we tried to get a hint from the human fMRI data by examining whether V1 BOLD activity differs between the EEG-defined states of burst-suppression and continuous slow waves. We focused on 12 human subjects, in which the EEG showed clear burst-suppression at the high sevoflurane concentration (3.9–4.6%) and continuous slow waves at both the intermediate (3%) and the low (2%) concentrations. For each of V1 and cingulate seeds, we examined the effects of sevoflurane concentration on BOLD signal amplitude and PSD. The detailed results are presented in *Figure 6—figure supplement 1*. In summary, both V1 and cingulate exhibited a reduction in BOLD signal amplitude from low to intermediate concentrations. During the high concentration (burst-suppression), the signal amplitude rose significantly in cingulate, but stayed the same as during the intermediate concentration in V1—albeit with a slight shift toward lower frequencies. Assuming the presence of continuous slow waves in V1 during the intermediate concentration, they are likely sustained through the high concentration as well. In case of a complete suppression of slow-wave activity, a noticeable further reduction in BOLD signal amplitude would be expected. A local change in neurovascular coupling, though unlikely, cannot be ruled out with the current data.

## Discussion

In this study, we identified the fMRI signatures of anesthesia-induced burst-suppression (*Figure 1*) and constructed comparable maps in four mammalian species: humans (*Figure 2*), macaques (*Figure 3*), marmosets (*Figure 4*), and rats (*Figure 5*). The maps exhibited homology across the three primate species; burst-suppression engaged the striatum and a large part of the neocortex, with some exceptions mostly at unimodal primary areas. The primary visual area (V1) was the predominant exception, appearing entirely decoupled from the rest of the cortex (*Figure 6*). Contrary to primates, burst-suppression was pancortical in rats, including all sensory and motor areas. In the following discussion, we first consider the limitations and potential applications of our method for identifying the fMRI signatures of burst-suppression, and then explore the implications of inter-species differences for existing models of burst-suppression and for future translational efforts.

### fMRI signatures of burst-suppression

We determined that EEG-defined burst-suppression in humans manifests on fMRI as an 'asymmetric PC,' which can be used as this state's fMRI signature (*Figure 1*). The asymmetry refers to the distribution of correlation coefficients between the PC and cortical voxels and reflects profound cortical synchrony. The threshold for PC asymmetry (median cortex-wide $r$) is arbitrarily defined, which constitutes a weakness, since the appropriate value is expected to vary depending on animal species, anesthetic agent, and imaging parameters. To create the animal burst-suppression maps, we conservatively

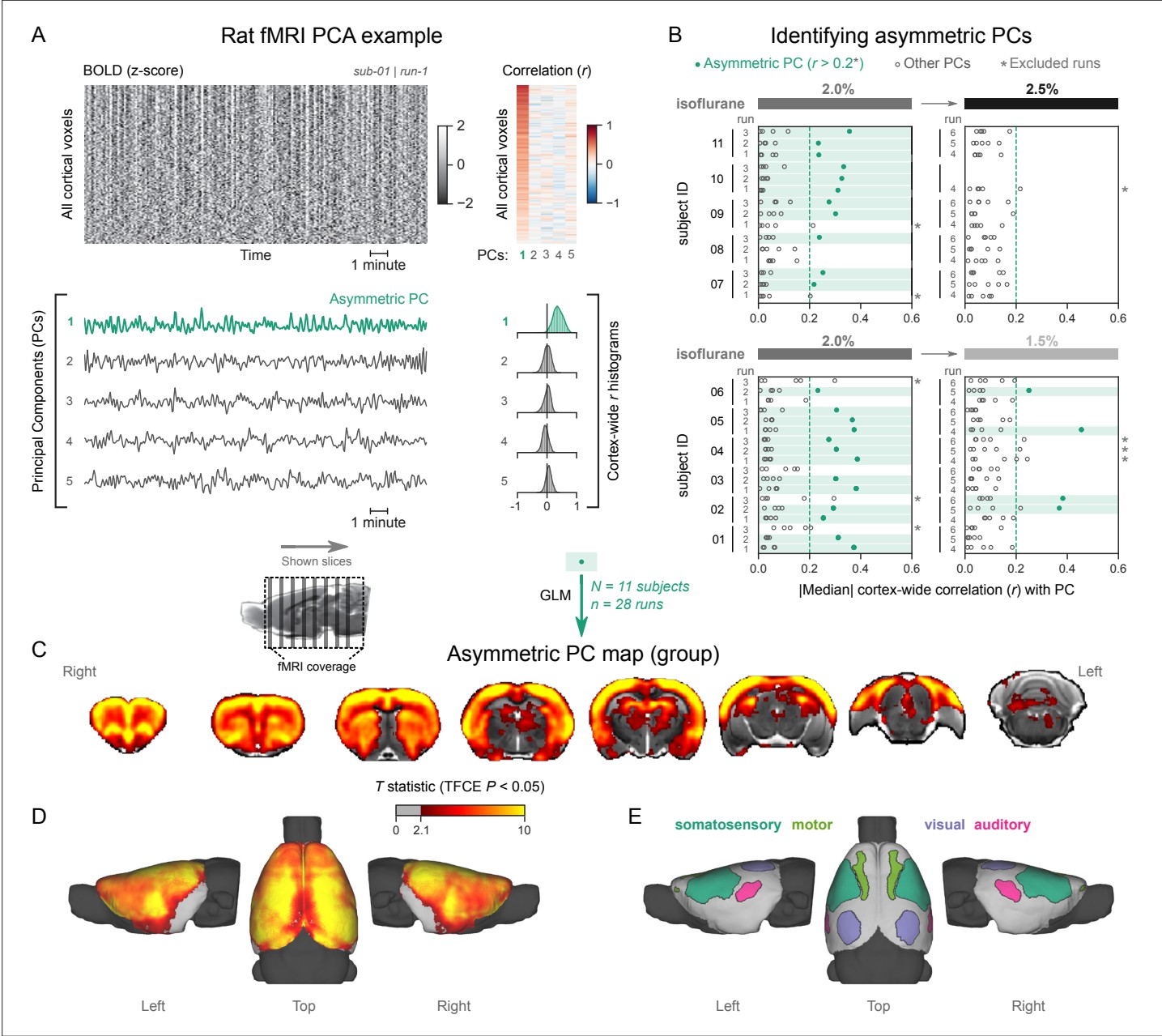

**Figure 5.** Rats exhibit pancortical functional magnetic resonance imaging (fMRI) signatures of burst-suppression. (**A**) The cortical blood-oxygen-level-dependent (BOLD) fMRI signal of a rat during isoflurane (2%) anesthesia is represented as a carpet plot, with rows (voxels) ordered according to their correlation with the mean cortical signal. The first five temporal principal components (PCs) of the signal are plotted below. The Pearson's correlation coefficients (*r*) between the PCs and all cortical voxels are represented both as a heatmap and as histograms for each PC. The first PC captures the widespread fluctuation visible on the carpet plot and has an asymmetric *r* histogram. *Figure 5—figure supplement 1* shows a counterexample, an fMRI run that exhibits no asymmetric PCs. (**B**) Cortex-wide median *r* values for the first five PCs are plotted as dots across the entire rat dataset (see *Figure 5—source data 1*). fMRI runs with a single prominent asymmetric PC (*r*>0.2, highlighted in green) are selected, excluding runs that had a second PC within *r*=0.15 of the most asymmetric PC (marked with an asterisk). The selected 28 runs serve as inputs for general linear model (GLM) analysis. (**C**) The group asymmetric PC map, computed via a second-level analysis of single-subject GLMs, is shown here overlaid on a study-specific volumetric template. The group statistics were carried out with FSL randomise; the resulting *T* statistic maps were thresholded using threshold-free cluster enhancement (TFCE) and a corrected p<0.05. *Figure 5—figure supplement 2* provides a closer look at subcortical structures. (**D**) The same group map is shown on a cortical surface representation of the template. The cerebellum and the olfactory bulb are shown in dark gray. (**E**) The locations of primary motor, somatosensory, auditory, and visual cortices—based on the SIGMA rat brain atlas—are indicated on the surface. *Figure 5—figure supplement 3* shows the unthresholded group asymmetric PC map in both volumetric and surface representations. *Figure 5—figure supplement 4* shows a temporal

*Figure 5 continued on next page*

*Figure 5 continued*

signal-to-noise ratio map overlaid on the volumetric template. In *Figure 5—figure supplement 5*, the asymmetric PC map is reproduced in a second rat fMRI dataset (Rat 2) from a previously published study.

The online version of this article includes the following source data and figure supplement(s) for figure 5:

**Source data 1.** Numerical data for *Figure 5B*.

**Figure supplement 1.** A functional magnetic resonance imaging (fMRI) run without asymmetric principal components (PCs).

**Figure supplement 2.** A closer look at subcortical structures.

**Figure supplement 3.** Unthresholded asymmetric principal component (PC) map.

**Figure supplement 4.** Temporal signal-to-noise ratio (tSNR) map.

**Figure supplement 5.** Identifying and mapping asymmetric principal components (PCs) in the Rat 2 dataset.

selected only fMRI runs that presented a single clear asymmetric PC candidate (see *Figures 3–5*). We have several reasons to be confident in our selection, despite the manually chosen thresholds. First, the isoflurane concentration range at which these asymmetric PCs were detected closely matches previous reports of burst-suppression in macaques (*Vincent et al., 2007*; *Zhang et al., 2019*) and rats (*Detsch et al., 2002*; *Liu et al., 2011*; *Masamoto et al., 2009*; *Stenroos et al., 2021*). In this regard, our findings in rats are especially encouraging, because almost all asymmetric PCs were found at a concentration of 2% and less so at 1.5 or 2.5% (*Figure 5*). Second, the detected widespread fluctuations cannot be attributed to known non-neural sources, such as physiology or motion. The asymmetric PCs displayed quasi-periodicity and were overwhelmingly correlated with gray matter areas. Physiological noise related to mechanical ventilation or heartbeat would have been strictly periodic, while motion artifacts are known to distribute along image edges. In summary, all evidence points toward burst-suppression as the underlying cause of the selected asymmetric PCs in the investigated animals.

Given the arbitrary threshold limitation, our method should not be viewed as an automated fMRI-based classification of burst-suppression but instead as a heuristic algorithm for identifying this state. Nevertheless, the method can find applications within the growing animal fMRI community (*Mandino et al., 2020*; *Milham et al., 2020*) and potentially also in clinical settings. Animal fMRI researchers could use the algorithm to screen datasets for the presence of burst-suppression—as we did in *Figure 5—figure supplement 5*. That said, the generalization of the method to other animal species and anesthetic agents warrants caution, keeping in mind that the amplitude and duration of bursts vary across anesthetics (*Akrawi et al., 1996*; *Fleischmann et al., 2018*). One potential clinical application concerns the management of comatose patients, where burst-suppression is often present—either transiently or throughout (*Brenner, 1985*; *Brown et al., 2010*; *Cloostermans et al., 2012*; *Hofmeijer et al., 2014*; *Young, 2000*). After appropriate validation, our method could map burst-suppression in these patients, even at centers lacking the capacity for simultaneous EEG-fMRI recordings. An fMRI readout of burst-suppression, with its high spatial resolution and whole-brain coverage, could provide clinicians with information inaccessible by EEG. The implications of this additional information for the management of comatose patients remain to be examined.

## Burst-suppression across mammalian species

How do our maps fit in what we know about the spatial distribution of burst-suppression? The apparent exclusion of V1 replicates the previous analysis in humans (*Golkowski et al., 2017*) and corroborates an existing report in macaques (*Zhang et al., 2019*). Still, our macaque map constitutes an important generalization, as it is based on seven long-tailed macaques—compared with only two rhesus individuals in the previous study. The marmoset burst-suppression map is, to our knowledge, the first of its kind and suggests that V1 exclusion may be a general primate feature. The pancortical distribution of burst-suppression in rats agrees with all previous findings in this species by fMRI (*Aedo-Jury et al., 2020*; *Liu et al., 2013*; *Liu et al., 2011*; *Paasonen et al., 2020*; *Schwalm et al., 2017*) or calcium imaging (*Ming et al., 2020*). The strength of our approach lies in computing all maps with the same data-driven approach, thus allowing us to make direct cross-species comparisons.

The main inter-species difference was the exclusion of visual areas (and parts of other unimodal cortices) in primates, but not in rats. While bursts were traditionally viewed as synchronous cortex-wide events, recent work rather describes them as complex waves (*Bojak et al., 2015*) emerging

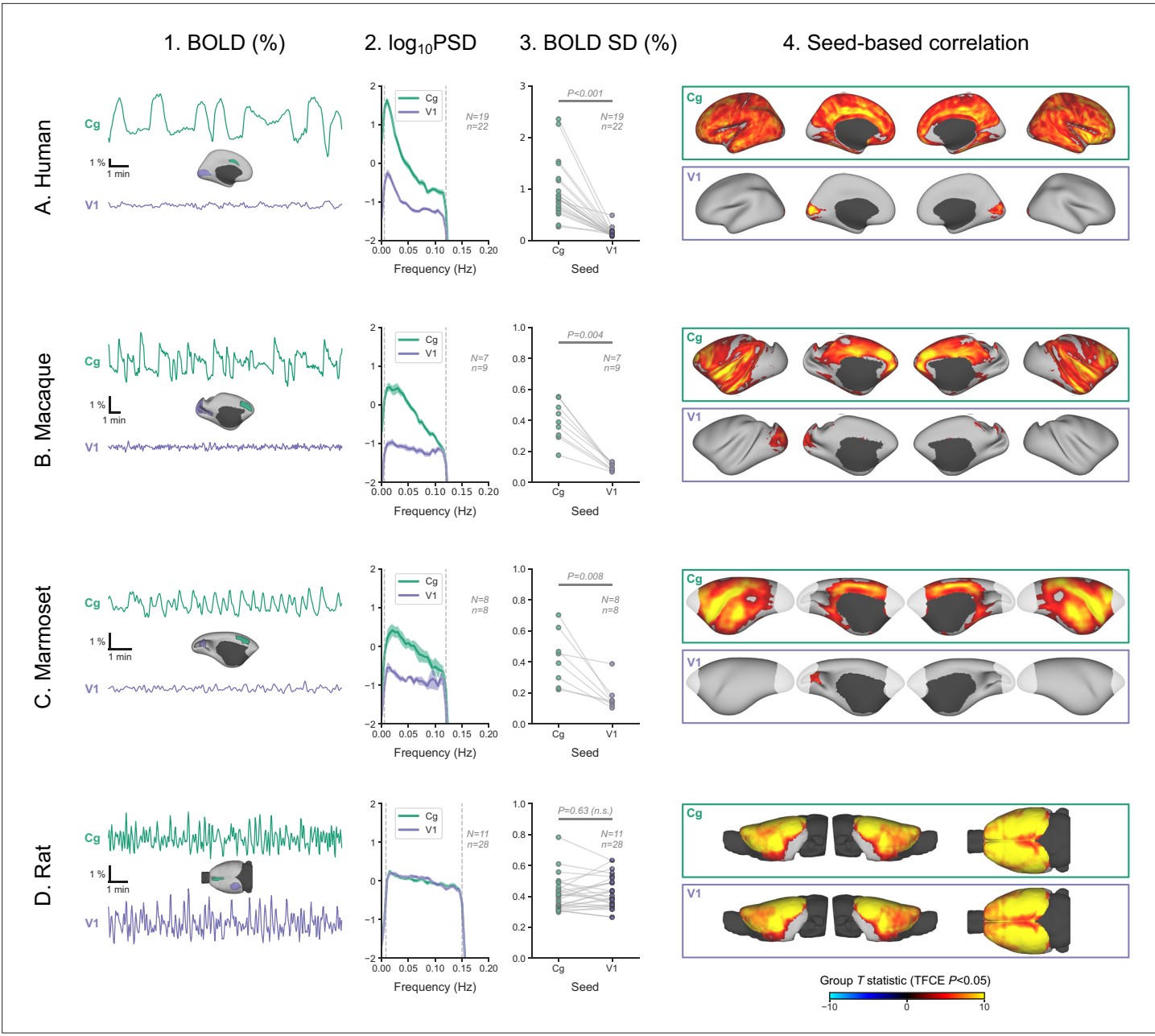

**Figure 6.** Primate V1 is uncoupled from the rest of the cortex during burst-suppression. (**A**) Blood-oxygen-level-dependent (BOLD) signal time series extracted from two regions-of-interest, the cingulate (Cg) area 24cd and the primary visual cortex (V1), are shown for an example human subject during burst-suppression (**A1**). The power spectral density (PSD) of the two regions' time series is plotted as mean ± SEM across all functional magnetic imaging (fMRI) runs with burst-suppression (**A2**). The SD of the time series is plotted for the same fMRI runs as a measure of BOLD signal amplitude (**A3**). The SDs of the two regions are compared using a paired samples two-tailed Wilcoxon rank-sum test (p values given). The results of seed-based correlation analysis—performed for each of the two regions—are shown overlaid on the cortical surface (**A4**). The group statistics were carried out with FSL randomise; the resulting *T* statistic maps were thresholded using threshold-free cluster enhancement (TFCE) and a corrected p<0.05. The other panels show the exact same plots as in A, but for macaques (**B**), marmosets (**C**), and rats (**D**), respectively. The cingulate (Cg) seed corresponds to the macaque area 24c, the marmoset area 24, and the rat primary cingulate cortex. Marmoset brain areas not covered by the fMRI volume are shown in white. The fMRI runs included in the analysis are the ones with an asymmetric PC, same as in previous figures. N: number of subjects; n: number of fMRI runs. The BOLD SD and PSD values across species and regions-of-interest are provided in *Figure 6—source data 1*. *Figure 6—figure supplement 1* shows how the BOLD signal time series from the two regions vary across sevoflurane concentrations in anesthetized humans.

The online version of this article includes the following source data and figure supplement(s) for figure 6:

**Source data 1.** BOLD SD and PSD values across species and regions-of-interest (numerical data for *Figure 6 A2–D2 and A3–D3*).

*Figure 6 continued on next page*

*Figure 6 continued*

**Figure supplement 1.** Blood-oxygen-level-dependent (BOLD) signal time series across sevoflurane concentrations in humans.

**Figure supplement 1—source data 1.** Results of repeated-measures ANOVA and post-hoc pairwise tests for *Figure 6—figure supplement 1*.

from a spatially shifting focus and rapidly spreading through the cortex (*Ming et al., 2020*). Electrocortical recordings in propofol-anesthetized human patients revealed that some electrodes could display continuous activity while others engaged in burst-suppression (*Lewis et al., 2013*). Together with our results, this implies that brain areas may vary in their propensity for generating and/or propagating bursts. Regional metabolic differences could underlie such variability, according to a proposed neurophysiological-metabolic model of burst-suppression (*Ching et al., 2012*). This model posits that suppressions occur when metabolic resources can no longer sustain neuronal firing, whereas bursts constitute transient recoveries of activity—enabled by slow metabolic replenishment. The visual cortex of primates could impose special metabolic demands and thus find itself at an extreme end of the spectrum for burst propensity. Such special demands could arise from its many distinctive characteristics—including cytoarchitecture, neurotransmitter receptor expression (*Froudist-Walsh et al., 2021*), functional connectivity profile (*Margulies et al., 2016*), and cortical myelin density (*Van Essen et al., 2019*). We particularly suspect cortical myelin to play a role, considering that many areas excluded from our primate burst-suppression maps—V1, MT, primary motor, and somatosensory cortices—are among the richest in myelin (*Van Essen et al., 2019*). Conversely, rodents exhibit a more uniform myelin distribution (*Fulcher et al., 2019*), which could translate in a likewise uniform propensity for bursting.

Whatever the cause, the non-correlation of V1 and other primary sensory areas with burst-suppression challenges existing ideas about the origin of bursts. Studies in isoflurane-anesthetized humans, cats, rats, and mice showed that bursts can be evoked by incoming sensory stimuli, including flashes of light (*Hartikainen et al., 1995*; *Hudetz and Imas, 2007*; *Kroeger and Amzica, 2007*; *Land et al., 2012*). Consequently, burst-suppression has been described as a state of cortical hypersensitivity, during which even subliminal sensory stimuli can trigger a burst excitation (*Kroeger and Amzica, 2007*). In that case, bursts would be expected to originate at the stimulated sensory area and then spread throughout the cortex indiscriminately, like ripples on a pond (*Muller et al., 2018*; *Sanchez-Vives et al., 2017*; *Schwalm et al., 2017*; *Stroh et al., 2013*). The pancortical rodent map is fully compatible with this view, but the exclusion of sensory areas in primates seems paradoxical: why would the areas of origin not be a part of the ripple? A possible explanation is that the bursts captured in our data were internally generated or evoked through a non-excluded sensory modality (e.g., auditory areas in non-human primates). Nevertheless, a closer investigation of sensory-evoked bursts in primates is warranted—especially with visual stimuli.

So far, we have discussed cortical aspects of burst-suppression, but fMRI also offers access to subcortical structures. Among these, the striatum displayed the strongest correlation with burst-suppression across all four species—showing that the cortex does not act in isolation. Cortico-striatal loops may play an important—but yet unexplored—role in burst propagation. The thalamus, on the contrary, was only partly correlated with burst-suppression, and to a varying extent across species. Interestingly, anteromedial parts were more likely to be involved than posterolateral parts, possibly reflecting a differential role played by thalamic primary relay centers and higher-order nuclei, as previously hypothesized (*Ming et al., 2020*). Regional variations in thalamic activity could even shape cortical differences since the thalamus may be important for driving burst onsets (*Ming et al., 2020*; *Steriade et al., 1994*). Similar to the thalamus, the hippocampus and amygdala were partially correlated with burst-suppression. However, we refrain from concrete conclusions, since in primates, these two structures are located in low signal-to-noise areas and often suffer from susceptibility-induced distortions (see figure supplement 4 of *Figures 2–5*). The cerebellum also suffered from low signal-to-noise in our non-human primate data and was only partly captured in marmosets and rats. Nevertheless, the non-correlation of the cerebellar cortex with burst-suppression is likely a true effect, given its consistency across all species.

At this point, we wish to address a potentially confusing issue of terminology. We employ the term 'burst-suppression,' preferred by clinical anesthesiologists. It has been pointed out (*Ming et al., 2020*) that several rodent studies may refer to the same phenomenon as 'slow oscillation' (*Aedo-Jury et al., 2020*; *Sanchez-Vives et al., 2017*; *Schwalm et al., 2017*; *Stroh et al., 2013*)—meaning a<1 Hz

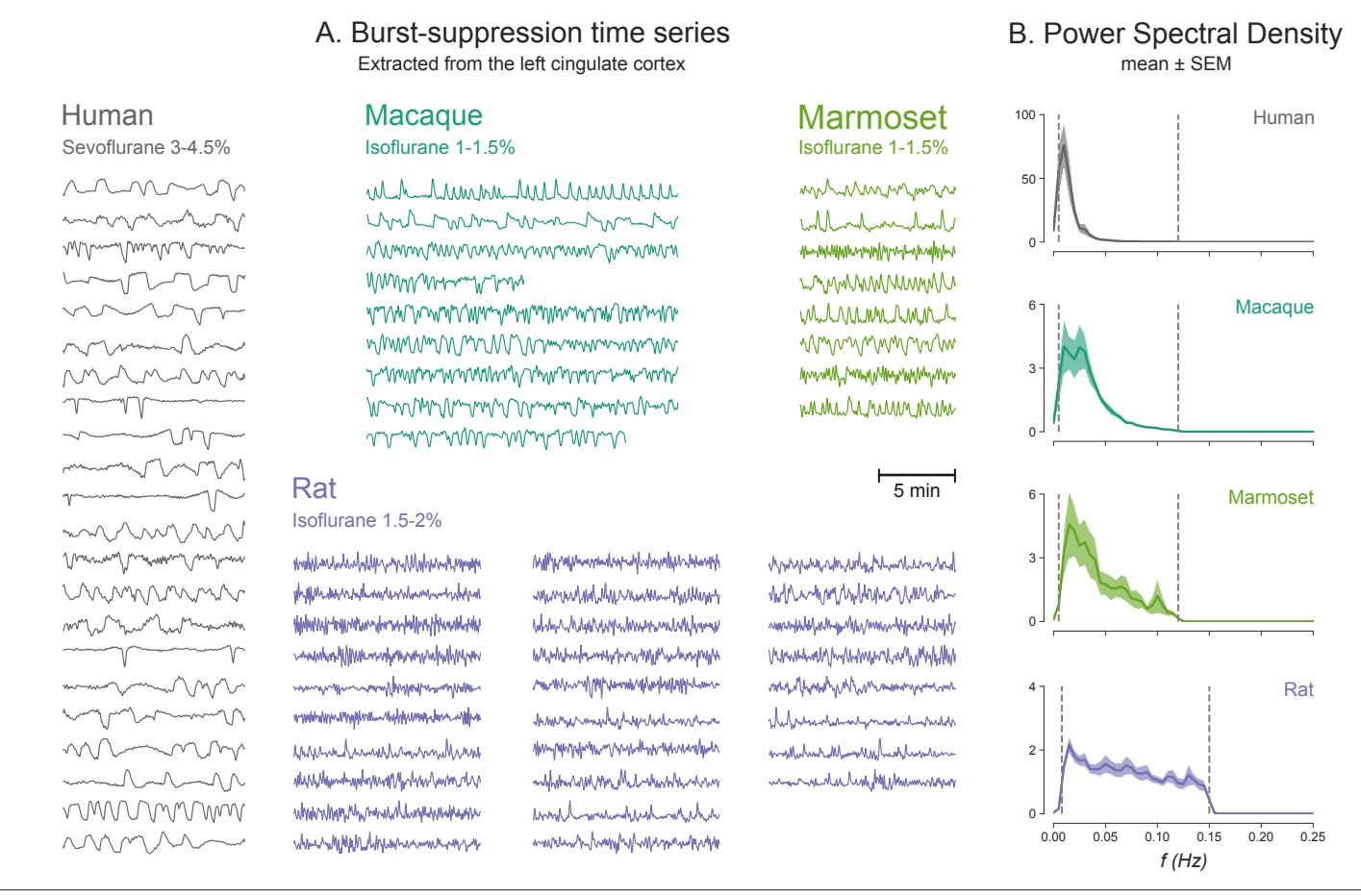

**Figure 7.** Burst-suppression timescales across species. (**A**) Blood-oxygen-level-dependent (BOLD) functional magnetic resonance imaging (fMRI) signal time series extracted from the cingulate cortex are shown for all runs classified as burst-suppression across the four species. This region—also used for the seed-based analysis (see *Figure 6*)—was selected because of its strong correlation with burst-suppression (i.e., asymmetric principal components) in all species. The time series are shown min-max scaled to emphasize differences in time, not in amplitude. The time series peaks appear shorter in duration and occur more frequently as we move from humans to non-human primates and finally to rats. (**B**) This effect is reflected on the power spectral density (PSD) plots of the time series, with increasingly higher frequencies being present from top to bottom. PSD was computed from time series normalized to percent-signal-change (relative to the mean) prior to min-max scaling. Vertical dashed lines indicate the cutoffs of the bandpass filter that was used for fMRI preprocessing.

alternation between a depolarized 'UP' state of neural activity and a hyperpolarized 'DOWN' state of silence (*Steriade and Nunez, 1993*). In the slow oscillation context, suppressions can be viewed as prolonged DOWN states (*Ching et al., 2012*; *Kroeger and Amzica, 2007*). However, the line between the two is blurry, given the gradual dose-dependent lengthening of quiescent periods. This distinction is even harder if one considers potential inter-species differences in the timescale of burst-suppression. We noticed that bursts and suppressions alternate faster when moving from humans to non-human primates to rats (*Figure 7*). Within the framework of the neurophysiological-metabolic model of burst-suppression (*Ching et al., 2012*), such differences in timescale could reflect the faster metabolic rate of smaller animals. Verifying this hypothesis would require further studies involving direct neural recordings.

A caveat concerning our cross-species comparison of burst-suppression maps and timescales might be that the depth of anesthesia was not standardized across species. For example, the three concentrations of sevoflurane delivered to human participants did not correspond to the three isoflurane levels used in rats. In humans, the highest sevoflurane concentration led to burst-suppression, with suppressions occupying roughly 50% of the time. In rats, burst-suppression was most reliably produced at the intermediate concentration of 2% isoflurane, while the 2.5% concentration probably corresponded to a near-constant isoelectric state. The non-human primate studies did not involve graded

**Table 1.** Datasets of the present study.

For age and weight, mean and range values are reported. Anesthetic concentration refers to the range used during functional magnetic resonance imaging acquisition. In humans and macaques the measured end-tidal concentration of the anesthetic gas is reported, in marmosets and rats the output concentration of the vaporizer.

| | Human | Macaque | Marmoset | Rat 1 | Rat 2 |
|---|---|---|---|---|---|
| Species/Strain | - | *M. fascicularis* | *C. jacchus* | *Wistar* | *Wistar* |
| Site | Munich | Göttingen | Göttingen | Göttingen | Kuopio |
| Field strength | 3T | 3T | 9.4T | 9.4T | 7T |
| Subjects (N) | 20 | 13 | 21 | 11 | 6 |
| Sex | M | F | 11 F | F | M |
| Age (years) | 26 (20–36) | 13.7 (6.8–19.8) | 6.0 (1.9–14.2) | - | - |
| Body Weight | - | 5.4 kg (3.6–8.1) | 407 g (337–517) | 398 g (350–450) | 307 g (265–350) |
| Anesthetic | Sevoflurane | Isoflurane | Isoflurane | Isoflurane | Isoflurane |
| Concentration (%) | 2.0–4.6 | 0.95–1.50 | 0.6–1.1 | 1.5–2.5 | 1.3 |

levels of anesthetic concentration; we rather analyzed the observed instances of burst-suppression. An additional complication is that our non-human primate studies covered a wide age range (see *Table 1*), extending well into old age. This is expected to influence the concentration at which burst-suppression occurs since anesthetic requirements decrease with age (*Nickalls and Mapleson, 2003*). In future studies, it would be of interest to systematically assess the dose-dependent evolution of burst-suppression across species.

Summing up, our findings in primates oppose the view of bursts as global cortical events and imply a varying propensity of brain regions to generate and/or propagate bursts. By contrast, burst-suppression in rats appears to be pancortical—prompting an investigation into the cause of this primate-rodent difference and raising questions about the validity of rodent models for anesthesia and coma. Addressing these issues will necessitate invasive neural recordings during burst-suppression, in both primates and rodents, with V1 being the target of highest interest. Our findings also point toward several other promising research directions: clarifying the exact relationship between spontaneous and sensory-evoked bursts, investigating the role of the striatum and thalamus, and examining inter-species differences in the timescale of burst-suppression.

## Materials and methods
### Experimental subjects and anesthesia

Five datasets from three research sites were used for this manuscript (see *Table 1*). The human data were obtained from a previous study and have been described in three other articles (*Golkowski et al., 2019*; *Golkowski et al., 2017*; *Ranft et al., 2016*). Experimental details for the second rat dataset (Rat 2) have also been presented elsewhere (*Paasonen et al., 2018*; *Paasonen et al., 2016*). All other experiments were carried out at the German Primate Center (Deutsches Primatenzentrum GmbH, Göttingen, Germany) with the approval of the ethics committee of the Lower Saxony State Office for Consumer Protection and Food Safety and in accordance with the guidelines from Directive 2010/63/EU of the European Parliament on the protection of animals used for scientific purposes. The relevant approval numbers are: 33.19-42502-04-16/2278 (Macaque), 33.19-42502-04-17/2496 (Marmoset), and 33.19-42502-04-15/2042 (Rat 1). One marmoset (sub-21) belonged to a different study with the approval number 33.19-42502-04-17/2535. All animals were purpose-bred, raised, and kept in accordance with the German Animal Welfare Act and the high institutional standards of the German Primate Center.

#### Human

The human EEG-fMRI data were acquired in 20 healthy adult men aged 20–36 years (mean 26). Anesthesia was induced by delivering a gradually increasing concentration of sevoflurane through

a tight-fitting face mask. When clinically indicated, a laryngeal mask was inserted and mechanical ventilation was started. Following that, sevoflurane concentration was increased until the EEG showed suppression about 50% of the time, with suppression periods lasting at least 1 s (reached at end-tidal concentrations of 3.9–4.6%). At this point, 700 s of simultaneous EEG-fMRI were recorded. This recording was repeated at two lower fixed concentrations, 3 and 2%, each following an equilibration period of 15 min. The lowest concentration recording was only acquired in 15/20 participants.

## Macaque

The macaque study included 15 female long-tailed macaques (*Macaca fascicularis*) in total, but fMRI data of sufficient duration (minimum 10 min) and quality were obtained only in 13/15. The macaques varied widely in age (6.8–19.8 years, mean 13.7) and weight (3.6–8.1 kg, mean 5.4). Prior to the imaging session, the animals were given no food for 6 hr and no water for 2 hr. Anesthesia was induced with an intramuscular (i.m.) injection of ketamine (3–10 mg/kg, Ketamin 10%, Medistar) and medetomidine (0.015–0.03 mg/kg, Dorbene vet 1 mg/ml, Zoetis). The animals were intubated using an endotracheal tube and placed in a sphinx position inside a custom-made MRI-compatible stereo-taxic frame to minimize movement artifacts. The ear bars also served as hearing protection and were dabbed with a lidocaine-containing ointment (EMLA 5%, AstraZeneca) for local anesthesia. Dexpan-thenol eye ointment (Bepanthen, Bayer) was applied for corneal protection. Body temperature was maintained constant at 37°C with the help of a circulating warm water system. Monitoring equipment was attached, including a pulse oximeter (NONIN model 7500FO, SANIMED GmbH, Germany), a respiratory belt, surface electrodes for electrocardiogram (MRI ECG electrodes, CONMED, USA), and an anesthetic gas analyzer (Philips M1026A Anesthesia Gas Module) for measuring $CO_2$ and isoflurane concentrations. Throughout the imaging session, the animals were mechanically ventilated with a respirator (Siemens-Elema AB SV 900C Servo Ventilator) at a rate of 8–14 bpm, using a mixture of medical air/$O_2$ (1:1 ratio) as the carrier gas. Anesthesia was maintained with isoflurane, with the concentration being continually adjusted to ensure physiological stability. The goal was to sustain an anesthetized depth that would ensure endotracheal tube toleration. The needed end-tidal isoflurane concentrations varied between 0.9 and 2% across animals. Each imaging session lasted for up to 5 hr, during which multiple structural MRI contrasts were acquired, as well as task-free BOLD fMRI (one or two runs per animal, each lasting 600–1200 s). During fMRI acquisition, end-tidal isoflurane concentra-tions ranged 0.95–1.5% across animals. At the end of the imaging session, isoflurane was stopped and the macaques were extubated as soon as spontaneous breathing was safely established.

## Marmoset

The marmoset study included a total of 36 common marmosets (*Callithrix jacchus*) of both sexes, but BOLD fMRI data was only obtained in 21 of those (11 females). The animals varied widely in age (1.9–14.2 years, mean 6) and weight (337–517 g, mean 407). Anesthesia was induced with a mixture of alfaxalone (12 mg/kg) and diazepam (3 mg/kg) injected i.m. This was followed by 0.05 ml glycopy-rronium bromide per animal (Robinul 0.2 mg/ml, Riemser Biosyn) to prevent secretions, maropitant (1 mg/kg, Cerenia, Pfizer) as an antiemetic, and meloxicam (0.2 mg/kg, Metacam, Boehringer Ingel-heim) as an anti-inflammatory analgesic. The animals were intubated using a custom-made flexible endotracheal tube and mechanically ventilated at a rate of 35–37 bpm (Animal Respirator Advanced 4601–2; TSE Systems GmbH, Bad Homburg, Germany). The marmosets were placed in a prone posi-tion inside a custom-built MRI-compatible bed equipped with a bite bar and ear bars. The ear bars also served as hearing protection and were dabbed with a lidocaine-containing ointment (EMLA 5%, AstraZeneca) for local anesthesia. Dexpanthenol eye ointment (Bepanthen, Bayer) was applied for corneal protection. Monitoring equipment consisted of a rectal temperature probe, a pneumatic pres-sure sensor placed on the chest, and three surface electrodes for ECG (MR-compatible Model 1030 monitoring and gating system; Small Animal Instruments Inc, Stony Brook, NY 11790, USA). Rectal temperature was kept within 36.5 ± 1°C using a pad heated by circulating water. Anesthesia was maintained with isoflurane delivered via the respirator, using a mixture of medical air/$O_2$ (1:1 ratio) as the carrier gas. The isoflurane concentration was adjusted individually for each animal, ranging 0.6–1.1%, in order to maintain stable anesthesia and physiology. Each imaging session lasted for up to 5 hr, during which multiple structural MRI contrasts were acquired, as well as one task-free fMRI run lasting 600 s (in sub-21, four task-free fMRI runs were obtained instead, each lasting 300 s). At the end

of the session isoflurane was stopped, and the marmosets were extubated as soon as spontaneous breathing was safely established.

## Rat 1

The Rat 1 dataset included 11 female adult Wistar rats (Charles Rivers Laboratories, Sulzfeld, Germany), weighing 350–450 g (mean 398). The rats were group-housed in cages with environmental enrichment, at a 12/12 hr light/dark cycle, with 20–24°C temperature and 45–55% humidity. Water and standard chow were provided ad libitum. Anesthesia was induced in a chamber with 5% isoflurane in 100% $O_2$. After the loss of the righting reflex, isoflurane was reduced to 2–3% and delivered through a nose cone. The rats were then intubated using a custom-made flexible tracheal tube and mechanically ventilated at a rate of 30 bpm (Animal Respirator Advanced 4601–2; TSE Systems GmbH, Bad Homburg, Germany). Isoflurane was stabilized at 2%, using medical air/$O_2$ mixture (1:1 ratio) as the carrier gas. The animals were fixed inside a custom-built MRI-compatible rat holder in a supine position (*Sirmpilatze et al., 2019*) and dexpanthenol eye ointment (Bepanthen, Bayer) was applied for corneal protection. Monitoring equipment was attached, consisting of a rectal temperature probe, a pneumatic pressure sensor placed on the chest, and three subcutaneous needle electrodes for ECG (MR-compatible Model 1030 monitoring and gating system; Small Animal Instruments Inc, Stony Brook, NY 11790, USA). Rectal temperature was kept within 36.5 ± 1°C using a pad heated by circulating water. The rat was placed at the isocenter of the MRI system and structural MRI data were acquired. These were followed by three task-free fMRI runs, each 360 s long. After that, isoflurane was either decreased to 1.5% (6/11 animals) or increased to 2.5% (5/11 animals). Following an equilibration period of 30 min, three additional task-free fMRI runs were acquired (six runs per animal in total). In one rat, only four fMRI runs could be obtained due to an early interruption of the imaging session (disconnection of the endotracheal tube). At the end of the imaging session, isoflurane was stopped, and the rats were extubated as soon as spontaneous breathing had recovered.

## Rat 2

The six adult male Wistar rats included in the Rat 2 dataset (*Paasonen et al., 2018*; *Paasonen et al., 2016*) weighed 265–350 g (mean 307 g). Briefly, anesthesia was induced with isoflurane 5%, maintained at 2% during animal preparation, and at 1.3% during imaging. The rats were endotracheally intubated and mechanically ventilated, with $N_2$:$O_2$ 70:30 mixture being used as the carrier gas. The data analyzed for this study comprised a single 300 s run of task-free fMRI per animal.

## MRI acquisition

### Human

Human data were acquired on a 3T whole-body MRI system (Achieva Quasar Dual 3.0T 16CH, Amsterdam, Netherlands) with an 8-channel phased-array head coil. fMRI data were collected using a gradient-echo echo planar imaging (EPI) sequence with the following parameters: repetition time 2 s, echo time 30 ms, flip angle 75°, field of view 220 × 220 mm$^2$, matrix size 96 × 96 (in-plane resolution of 2.3 × 2.3 mm$^2$), 35 axial slices with 3 mm thickness and 1 mm interslice gap, and acquisition time 700 s (350 volumes). A 1 mm isotropic T1-weighted image was acquired before the fMRI runs to serve as each subject's anatomical reference.

### Macaque

Macaque data were acquired on a 3 Tesla whole-body MRI system (Siemens MAGNETOM Prisma 3T, Siemens Healthcare GmbH, Erlangen, Germany). The body coil was used for signal transmission and the 70 mm loop coil for reception. fMRI data were collected using a gradient-echo EPI sequence with the following parameters: repetition time 2 s, echo time 27 ms, flip angle 90°, field of view 96 × 96 mm$^2$, matrix size 80 × 80 (in-plane resolution of 1.2 × 1.2 mm$^2$), 33 contiguous axial slices with 1.2 mm thickness, acquisition time 600–1200 s (300–600 volumes). For anatomical reference, a 0.5 mm isotropic T1-weighted image was collected using the MPRAGE sequence with the following parameters: 2 averages, repetition time 2.7 s, inversion time 850 ms, echo time 2.76 ms, flip angle 8°, field of view 12.8 × 10.8 cm$^2$, matrix size 256 × 216, and 192 contiguous axial slices with 0.5 mm thickness.

## Marmoset

Marmoset data were acquired on a 9.4T Bruker BioSpec MRI system, equipped with the B-GA 20S gradient, and operated via ParaVision 6.0.1 software (Bruker BioSpin MRI GmbH, Ettlingen, Germany). Signal was transmitted with a volume resonator (inner diameter 154 mm, Bruker BioSpin MRI GmbH) and received with a 40 × 43 mm loop coil (Rapid Biomedical GmbH, Rimpar, Germany). A field map was measured and shims were adjusted to ensure homogeneity in an ellipsoidal volume within the marmoset brain (MAPSHIM). fMRI data were collected using a gradient-echo EPI sequence with the following parameters: repetition time 2 s, echo time 18 ms, flip angle 90°, field of view 62.4 × 25.6 $mm^2$, matrix size 156 × 64 (in-plane resolution of 0.4 × 0.4 $mm^2$), 30 contiguous coronal slices with 0.8 mm thickness, and acquisition time 600 s (300 volumes). The slices did not cover the entire brain in the rostro-caudal direction; they extended approximately from the anterior end of the corpus callosum to the middle of the cerebellum (see fMRI coverage in *Figure 4*). In one marmoset (sub-21), we achieved whole-brain coverage by acquiring 40 contiguous coronal slices. In this animal, four fMRI runs were acquired at rest, each lasting 300 s (150 volumes). For anatomical reference, we used a 0.21 mm isotropic magnetization transfer (MT)-weighted image, acquired using a 3D, RF-spoiled, fast low-angle shot sequence with the following parameters: 2 averages, repetition time 16.1 ms, echo time 3.8 ms, flip angle 5°, field of view 37.8 × 37.8 × 37.8 $mm^3$, and matrix size 180 × 180 × 180.

## Rat 1

Rat data were acquired on a 9.4T Bruker BioSpec MRI system, equipped with the B-GA 12S2 gradient, and operated via ParaVision 6.0.1 software (Bruker BioSpin MRI GmbH, Ettlingen, Germany). Signal was transmitted with a volume resonator (inner diameter 86 mm) and received with a rat brain 4-channel quadrature surface coil (both from Bruker BioSpin). A field map was measured and shims were adjusted to ensure homogeneity in an ellipsoidal volume within the rat brain (MAPSHIM). fMRI data were collected using a gradient-echo EPI sequence with the following parameters: repetition time 2 s, echo time 15 ms, flip angle 70°, field of view 25.6 × 19.2 $mm^2$, matrix size 128 × 96 (in-plane resolution of 0.2 × 0.2 $mm^2$), 40 contiguous coronal slices with 0.5 mm thickness, and acquisition time 720 s (360 volumes). The fMRI slices covered the entire rat brain, excluding the olfactory bulbs and the caudal 1/4 of the cerebellum (see fMRI coverage in *Figure 5*). For anatomical reference, a T2-weighted image was acquired using the TurboRARE (rapid acquisition with relaxation enhancement) sequence with the following parameters: 2 averages, RARE factor 8, repetition time 6.3 s, effective echo time 40 ms, flip angle 90°, field of view 32 × 32 $mm^2$, matrix size 256 × 256 (in-plane resolution 0.125 × 0.125 $mm^2$), and 50 contiguous coronal slices with 0.5 mm thickness.

## Rat 2

Data were acquired on a 7T Bruker PharmaScan MRI system and operated via ParaVision 5.1 software (Bruker BioSpin MRI GmbH, Ettlingen, Germany). Signal was transmitted with a volume resonator (inner diameter 72 mm) and received with a rat brain 4-channel quadrature surface coil (both from Bruker BioSpin). fMRI data were collected using a spin-echo EPI sequence with the following parameters: repetition time 2 s, echo time 45 ms, field of view 25 × 25 $mm^2$, matrix size 64 × 64 (in-plane resolution of 0.39 × 0.39 $mm^2$), 9 contiguous coronal slices with 1.5 mm thickness, and acquisition time 600 s (300 volumes). The fMRI slices were placed from the posterior end of the olfactory bulb to the anterior end of the cerebellum (see fMRI coverage in *Figure 5—figure supplement 5*).

## MRI preprocessing

To facilitate inter-species comparability, we aimed for harmonizing preprocessing across datasets. The fMRI preprocessing steps were kept the same, but their parameters were adapted to each dataset. We used functions from multiple freely available neuroimaging toolkits, including ANTs v2.1.0 (Advanced Normalization Tools, *Avants et al., 2011*), FSL v5.0.1 (FMRIB Software Library, *Jenkinson et al., 2012*), AFNI v18.2.06 (*Cox, 1996*), Freesurfer v6.0.0 (*Fischl, 2012*), and Connectome Workbench v1.4.2 (*Marcus et al., 2011*). Functions from these toolkits were combined into pipelines using the Nipype (v1.2) framework (*Gorgolewski et al., 2011*). The exact steps and functions used will be presented in detail for each dataset. We will also describe the species-specific standard coordinate spaces (templates) and atlases (parcellations) that were used for group analysis and data visualization.

## Templates and atlases

For human data, we used the MNI152 template (MNI-ICBM average of 152 T1-weighted MRI scans, nonlinear sixth generation, distributed with FSL v5.0.1, *Grabner et al., 2006*) as the standard coordinate system. For visualization purposes, results were resampled from the MNI152 volumetric space to the fsaverage surface template (*Fischl et al., 1999*, part of Freesurfer v6.0.0) using the registration-fusion approach (RF-ANTS, *Wu et al., 2018*). Cortical regions-of-interest were taken from the Human Connectome Project multimodal parcellation 1.0 (*Glasser et al., 2016*) and transformed to the MNI152 and fsaverage spaces. Subcortical regions were defined based on the Harvard-Oxford subcortical structural atlas (distributed with FSL v5.0.1). For marmoset data, we used the NIH v3.0 population template (*Liu et al., 2021*) as a standard coordinate system. The volumetric results were visualized on the template's cortical surface (Connectome Workbench *volume-to-surface-mapping*, ribbon-constrained method). Regions-of-interest were taken from the Marmoset Brain Mapping V1 MRI-based parcellation (*Liu et al., 2018*). For the macaque and rat data, we built our own study-specific volumetric templates using a validated template creation process implemented in ANTs (*Avants et al., 2010*). This process relies on iterative nonlinear registration to produce an unbiased average of the population. Our workflow was constructed similarly to the NMT template of the rhesus macaque (*Seidlitz et al., 2018*). For each species, the anatomical images (T1-weighted for macaques, T2-weighted for rats) were corrected for variations in image intensity (ANTs *N4BiasFieldCorrection*, *Tustison et al., 2010*) and rigidly aligned using ANTs (*antsRegistration*). The voxel-wise average of the rigidly aligned images served as the initial target for the iterative template creation process (ANTs *buil dtemplateparallel.sh*). The resulting template images were segmented semi-manually with ITK-SNAP (*Yushkevich et al., 2006*) to create masks for brain and cortex. Macaque regions-of-interest were defined based on the cortical hierarchical (CHARM, *Jung et al., 2021*) and subcortical (SARM, *Hartig et al., 2021*) atlases of the rhesus macaque, which were transformed from the NMT V2 space to our macaque template space. Rat regions-of-interest were based on the SIGMA rat brain atlas (*Barrière et al., 2019*) and were transformed from the SIGMA space. We reconstructed the study-specific macaque template's cortical surface (white matter and pial) using the *precon_all* pipeline (*Benn, 2022*) and resampled the volumetric results to it (Connectome Workbench *volume-to-surface-mapping*, ribbon-constrained method). The rat brain surface was reconstructed by extracting the isosurface of the brain mask (Freesurfer) and smoothing it (Connectome Workbench). For representing volumetric results on the rat cortical surface, we adapted a method previously used for the mouse brain (*Huntenburg et al., 2021*). In short, we first computed the surface normals, vectors perpendicular to each face of the surface mesh. We then generated line segments along those normals, with each segment stretching from the inner edge of the cortical mask toward the pial surface. Finally, we averaged the cortical voxels overlapping with each of these line segments and assigned their mean value to the corresponding surface point.

## Human

Images were converted from PAR/REC to NIfTI (Neuroimaging Informatics Technology Initiative) format using *dcm2niix* v1.0 (*Li et al., 2016*). Structural T1-weigthed images were processed using Freesurfer's *recon-all* command, which includes brain extraction and segmentation. The extracted brains were registered to the MNI152 template using linear transforms (*antsRegistration* affine, 12 degrees of freedom), followed by nonlinear symmetric diffeomorphic registration (*antsRegistration* SyN, *Avants et al., 2008*). The fMRI time series were corrected for slice-timing (FSL *slicetimer*) and motion (FSL *MCFLIRT*), and the brain was extracted (FSL *BET*). Polynomial trends were removed up to the third degree, the time series were bandpass filtered at 0.005–0.12 Hz, and spatially smoothed using a 3D Gaussian kernel with full-width-at-half-maximum (FWHM) of 4 mm (AFNI *3dTProject*). A rigid transformation matrix was calculated between the mean brain-extracted fMRI volume and the brain-extracted structural image of the same subject (*antsRegistration*). The matrix was combined with the previously calculated structural-to-template transforms into a composite warp file, which was used to resample the preprocessed fMRI time series into the template space, at 2 mm isotropic resolution (*antsApplyTransforms*).

## Macaque

Images were converted from DICOM to NIfTI format (*dcm2niix*) and rotated from the sphinx into standard orientation. Structural T1-weighted images were corrected for intensity inhomogeneities (ANTs *N4BiasFieldCorrection*) and registered to the study-specific macaque template (*antsRegistration* affine, followed by SyN). The template's brain mask was transformed back to the single-subject structural space, and the brain was extracted. The fMRI time series were corrected for slice-timing (FSL *slicetimer*) and motion (FSL *MCFLIRT*), and the brain was extracted semi-manually with ITK-SNAP. Polynomial trends were removed up to the third degree, the time series were bandpass filtered at 0.005–0.12 Hz, and spatially smoothed using a 3D Gaussian kernel with FWHM of 2 mm (AFNI *3dTProject*). The mean brain-extracted fMRI volume was rigidly registered to the same subject's structural brain image (*antsRegistration*). The resulting transformation matrix was combined with the previously calculated structural-to-template transforms into a composite warp file, which was used to resample the preprocessed fMRI time series into the template space, at 1 mm isotropic resolution (*antsApplyTransforms*).

## Marmoset

Images were exported from ParaVision 6.0.1 to DICOM format, converted from DICOM to NIfTI (*dcm2niix*), and rotated into standard orientation. Structural images (MT-weighted) were corrected for intensity inhomogeneities (ANTs *N4BiasFieldCorrection*) and registered to the T2-weighted image of the NIH v3.0 population template (*antsRegistration* affine, followed by SyN). The fMRI time series were corrected for slice-timing (FSL *slicetimer*) and motion (*antsMotionCorr*). Polynomial trends were removed up to the third degree, the time series were bandpass filtered at 0.005–0.12 Hz, and spatially smoothed using a 3D Gaussian kernel with FWHM of 1 mm (AFNI *3dTProject*). The mean fMRI volume was rigidly registered to the subject's structural image (*antsRegistration*). The resulting transformation matrix was combined with the previously calculated structural-to-template transforms into a composite warp file, which was used to resample the preprocessed fMRI time series into the template space, at 0.4 mm isotropic resolution (*antsApplyTransforms*).

## Rat

The rat preprocessing pipeline was identical to the one used for marmosets, with a few differences. The registration target for structural T2-weighted images was our study-specific rat brain template. The 3D Gaussian kernel's FWHM was set to 0.6 mm, and the preprocessed fMRI time series were resampled into the template space at a resolution of $0.2 \times 0.2 \times 0.5$ mm$^3$. Bandpass temporal filtering was initially applied using the same cutoffs as for the other species (0.005–0.12 Hz). However, we later realized that bursting activity in rats also occurred at faster timescales (see *Figure 7*) and reanalyzed the data using higher bandpass cutoffs (0.008–0.15 Hz). This gave a richer representation of burst-suppression in the temporal domain, without qualitatively altering the resulting maps.

## Carpet plots and PCA

This part of the analysis was performed using in-house python code, which we have made publicly available through GitHub (*Sirmpilatze, 2021*) and archived on Zenodo (doi: 10.5281/zenodo.5545695). We imported the preprocessed fMRI time series (in their native space) into python and computed the mean and SD images across time as well as their ratio (tSNR). Voxels with tSNR <15 were omitted from further analysis. We then applied cortical masks, which we transformed from the species-specific templates to the native fMRI space and retained only cortical voxels. Subsequently, we reshaped the data from 4D to a 2D matrix (voxels × frames), with each voxel time series (row) normalized to zero mean and unit variance (z-score). To make global signal fluctuations more apparent on the 2D carpet plot, we reordered the voxels according to decreasing correlation with the mean cortical time series (*Aquino et al., 2020*). We then performed PCA on the carpet matrix using singular value decomposition. We extracted the first five temporal PCs and correlated them with all retained cortical voxels (Pearson's *r*). If a PCs' cortex-wide median *r* value was negative, we sign-flipped the PC and its correlation values to force the PC polarity to match the polarity of BOLD signal fluctuations for most voxels. We visualized the correlation values as heatmaps and as histograms per PC (50 bins between –1 and 1). The PCs with cortex-wide median *r* value above a certain species-specific threshold (human 0.45, macaque 0.15, marmoset 0.22, and rat 0.2) were classified as 'asymmetric PCs'. The threshold

was selected so that at most one prominent PC would exceed them for each fMRI run, with other PCs far below the threshold (see *Figure 1* and *Figures 3–5*). A few fMRI runs in the Rat 1 dataset had more than one PC close to or above the threshold and were thus excluded from downstream analysis (*Figure 5B*).

## Correspondence to EEG

The human EEG acquisition and preprocessing has been described in past manuscripts (*Golkowski et al., 2017*; *Ranft et al., 2016*). To identify burst and suppression epochs, we calculated the PSD of frontopolar electrodes using Welch's method (6 s Hann windows, each centered on a single fMRI volume). fMRI volumes with low EEG power (<100 µV$^2$) in the 0.5–8 Hz band were designated as suppressions and assigned value 0. Non-suppression epochs, which displayed high amplitude activity mainly in the delta band, were designated as bursts and assigned value 1 (see example in *Figure 1A*). This process was performed for all anesthetized EEG-fMRI recordings—not just for the ones acquired with the high sevoflurane concentration (as in the previous analysis, *Golkowski et al., 2017*), which led us to discover that four intermediate-concentration recordings also contained suppression epochs. Recordings in which no fMRI volume was designated as suppression were classified as being in a continuous slow-wave state.

For runs defined as burst-suppression by EEG, we convolved the boxcar envelope (bursts 1, and suppression 0) with a canonical two-gamma HRF (*Glover, 1999*) and applied the same bandpass filter as for fMRI preprocessing (0.005–0.12 Hz, see *Figure 1B*). To compare the resulting EEG-derived hemodynamic model of burst-suppression with the asymmetric PC derived from the simultaneous fMRI recording, we computed their cross-correlation. We extracted each pair's correlation at zero lag, their maximum cross-correlation, and the time lag at which the maximum was achieved. Two runs acquired with the high sevoflurane concentration were excluded from the cross-correlation analysis: subject 15 (EEG contained only one short burst and fMRI contained no asymmetric PC) and subject 7 (burst-suppression could be seen on EEG, but part of the recording was corrupted).

## Burst-suppression maps

For each fMRI run with an asymmetric PC, we used the PC as a regressor in a first-level GLM analysis to get its distribution across the brain. The GLM was carried out with FEAT (fMRI Expert Analysis Tool Version 6.00, part of FSL) using local autocorrelation correction (*Woolrich et al., 2001*). Resulting $Z$ statistic images were masked for the brain and thresholded non-parametrically using clusters determined by $Z>3.1$ and a corrected cluster significance threshold of p=0.05 (*Worsley, 2001*). In the case of the human dataset, we also conducted the same analysis using the EEG-derived hemodynamic models as regressors. We compared the unthresholded $Z$ statistic maps resulting from the two analyses by computing their spatial cross-correlation (ANTs *Measure Image Similarity*, neighborhood cross-correlation metric within the brain mask). We wish to emphasize that no nuisance signals (e.g., global or CSF signal) were regressed out or included as covariates in the GLM. Since the burst-suppression fluctuation is very widespread, it is expected to be a major contributor in the global signal and hence be highly correlated with it. Similarly, the CSF signal is also related to burst-suppression activity (see *Figure 2—figure supplement 2*) and cannot be considered a nuisance variable.

For each of the four species, we computed group-level burst-suppression maps using the resulting parameter estimate images from first-level GLMs. For subjects having more than one fMRI run with asymmetric PCs, the images were first averaged within each subject. Subsequently, a non-parametric one sample T-test was computed on the list of subject-level images using FSL randomise (*Winkler et al., 2014*) with TFCE (*Smith and Nichols, 2009*). The permutation-based randomise algorithm allows for control of false positives with minimal assumptions about the data, while TFCE has the sensitivity benefits of cluster-based thresholding—without the need for arbitrary decisions on cluster-forming thresholds. We thought these advantages to be important, given our goal to compare maps across dissimilar datasets, with varying spatial scales and signal-to-noise ratios. The resulting unthresholded group maps are given in figure supplement 3 of *Figures 2–5* across species. Areas with a corrected p<0.05 were judged to be significantly correlated with asymmetric PCs. Significant anticorrelations were computed by repeating the above procedure with a flipped contrast. The thresholded maps are shown in *Figures 2–5*. All group statistics were carried out in the species-specific volumetric

template spaces. The resampling on cortical surfaces was done for purposes of data visualization only, using the methods described in the 'Template and atlases' paragraph.

The marmoset whole-brain map (shown in *Figure 4—figure supplement 5*) was computed via a fixed-effects analysis of the four fMRI runs acquired in sub-21. The *Z* (gaussianized *T/F*) statistic image was thresholded non-parametrically using gaussian random field theory-based maximum height thresholding with a (corrected) significance threshold of p=0.05 (*Worsley, 2001*).

### Region-of-interest analysis

We defined two regions-of-interest per species—one in the left V1 and another in the left cingulate cortex (Cg: human area 24 cd, macaque area 24 c, marmoset area 24, and rat primary cingulate cortex; see *Figure 6*). The former was chosen because of its exclusion from primate burst-suppression maps; the latter because of its strong correlation with burst-suppression across species. We extracted each region's average BOLD signal time series from all preprocessed fMRI runs with asymmetric PCs and normalized them as percent-signal-change relative to the temporal mean. We quantified the SD of the normalized time series as a measure of BOLD signal amplitude. For each species, we compared the BOLD SD between the two regions using a paired samples two-sided Wilcoxon rank-sum test (jamovi v1.8 statistical software). We also computed the PSD of the normalized time series using Welch's method (as implemented in *scipy.signal.welch*, segments of 100 timepoints, with 50 overlapping points between segments). Finally, each normalized time series was used as a regressor in a seed-based GLM analysis. The GLM statistics and thresholding were carried out exactly as described above for burst-suppression maps. A total of eight seed-based group-level analyses were carried out (4 species × 2 seeds).

In the human dataset, we further examined whether the BOLD signal SD and PSD of the two regions differed significantly between the two EEG-defined states. We focused on 12 human subjects, in which the EEG showed clear burst-suppression at the high sevoflurane concentration (3.9–4.6%) and continuous slow waves at both the intermediate (3%) and the low (2%) sevoflurane concentrations. For each of V1 and cingulate seeds, we examined the effects of sevoflurane concentration on three response variables: BOLD signal amplitude (SD), low-frequency $\log_{10}$PSD (integrated between 0.005 and 0.05 Hz), and high-frequency $\log_{10}$PSD (integrated between 0.05 and 0.12 Hz). The lower frequency range corresponded to the timescale of burst-suppression in the human dataset. The effect of sevoflurane concentration on each of the three response variables was probed with repeated-measures ANOVA (sevoflurane concentration as the within-subject repeated-measures factor, jamovi v1.8). The Greenhouse-Geisser sphericity correction was applied when needed (Mauchly's sphericity test with p<0.05). In cases of significant ANOVA effects (p<0.05), we performed Tukey post-hoc pairwise comparisons between sevoflurane concentration levels (jamovi 1.8). Least-squares means were estimated for the three levels of sevoflurane concentration. Results are summarized in *Figure 6—figure supplement 1*.

## Acknowledgements

This work was supported by the DFG Center for Molecular Physiology of the Brain (DFG project number 5485646) and the Leibniz Science Campus Primate Cognition. We wish to thank: Kristin Kötz and Kerstin Fuhrmann for technical assistance; Dr. Iris Steinmann for advice concerning EEG data analysis; Dr. Myrto Panopoulou, Tor Rasmus Memhave, and Christina Kajba for proofreading.

## Additional information

### Funding

| Funder | Grant reference number | Author |
| --- | --- | --- |
| Deutsche Forschungsgemeinschaft | DFG Center for Molecular Physiology of the Brain 5485646 | Nikoloz Sirmpilatze Susann Boretius |

| Funder | Grant reference number | Author |
|---|---|---|
| Leibniz Science Campus Primate Cognition | | Susann Boretius |
| International Max Planck Research School for Neurosciences | open access funding | Nikoloz Sirmpilatze |

The funders had no role in study design, data collection and interpretation, or the decision to submit the work for publication.

## Author contributions

Nikoloz Sirmpilatze, Conceptualization, Data curation, Formal analysis, Investigation, Methodology, Project administration, Software, Validation, Visualization, Writing - original draft, Writing - review and editing; Judith Mylius, Investigation, Writing - review and editing; Michael Ortiz-Rios, Jürgen Baudewig, Validation, Writing - review and editing; Jaakko Paasonen, Daniel Golkowski, Data curation, Resources, Writing - review and editing; Andreas Ranft, Rüdiger Ilg, Olli Gröhn, Resources, Writing - review and editing; Susann Boretius, Conceptualization, Funding acquisition, Investigation, Methodology, Project administration, Supervision, Validation, Writing - review and editing

## Author ORCIDs

Nikoloz Sirmpilatze ⓘ http://orcid.org/0000-0003-1778-2427
Michael Ortiz-Rios ⓘ http://orcid.org/0000-0003-4862-654X
Olli Gröhn ⓘ http://orcid.org/0000-0003-1372-1651
Susann Boretius ⓘ http://orcid.org/0000-0003-2792-7423

## Ethics

The human data were acquired for a previous study and have been described elsewhere (Ranft et al., 2016), including information on participant recruitment and the exclusion criteria. The study had been approved by the ethics committee of the Medical School of the Technische Universität München (Munich, Germany).

Four animal datasets (Macaque, Marmoset, Rat 1, Rat 2) were used in this manuscript (see Table 1). The Rat 2 dataset was acquired for a previous study and has been described elsewhere (Paasonen et al., 2018, 2016). The relevant experiments had been approved by the Animal Ethics Committee of the Provincial Government of Southern Finland. The experiments for the three novel animal datasets were carried out at the German Primate Center (Deutsches Primatenzentrum GmbH, Göttingen, Germany) with the approval of the ethics committee of the Lower Saxony State Office for Consumer Protection and Food Safety and in accordance with the guidelines from Directive 2010/63/EU of the European Parliament on the protection of animals used for scientific purposes. The relevant approval numbers are: 33.19-42502-04-16/2278 (Macaque), 33.19-42502-04-17/2496 and 33.19-42502-04-17/2535 (Marmoset), 33.19-42502-04-15/2042 (Rat 1). All animals were purpose-bred, raised, and kept in accordance with the German Animal Welfare Act and the high institutional standards of the German Primate Center.

## Decision letter and Author response

Decision letter https://doi.org/10.7554/eLife.74813.sa1
Author response https://doi.org/10.7554/eLife.74813.sa2

# Additional files

## Supplementary files

• Transparent reporting form

## Data availability

Source Data files have been provided for Figure 1G-H, Figure 2A, Figure 3B, Figure 3-figure supplement 4B, Figure 4B, Figure 4-figure supplement 5B, Figure 5B, Figure 6, and Figure 6-figure supplement 1. The custom python code implementing the carpet plots and PCA analysis is publicly available through GitHub (Sirmpilatze, 2021) and Zenodo (doi:https://doi.org/10.5281/zenodo.5545695). All animal MRI datasets are openly available on Zenodo (doi:https://doi.org/10.5281/zenodo.5565305).

The following datasets were generated:

| Author(s) | Year | Dataset title | Dataset URL | Database and Identifier |
|---|---|---|---|---|
| Sirmpilatze N, Mylius J, Paasonen J, Gröhn O, Boretius S | 2021 | Functional MRI data from isoflurane-anesthetized macaques, marmosets, and rats | https://doi.org/10.5281/zenodo.5565305 | Zenodo, 10.5281/zenodo.5565305 |
| Sirmpilatze N | 2021 | niksirbi/pcarpet: Pre-release of pcarpet | https://doi.org/10.5281/zenodo.5545695 | Zenodo, 10.5281/zenodo.5545695 |

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
