## [Editor Report]

This study reveals that anesthesia-induced burst suppression's spatial patterns differ across humans, macaques, marmosets, and rats. Given that burst suppression is considered a hallmark of unconscious states, these findings are potentially important for us to understand the evolution of the neural correlates of consciousness.

---

## [Decision Letter]

**Decision letter after peer review:**

Thank you for submitting your article "Spatial signatures of anesthesia-induced burst-suppression differ between primates and rodents" for consideration by *eLife*. Your article has been reviewed by 3 peer reviewers, and the evaluation has been overseen by a Reviewing Editor and Tamar Makin as the Senior Editor. The following individuals involved in review of your submission have agreed to reveal their identity: Shuntaro Sasai (Reviewer #2); Cornelius Faber (Reviewer #3).

Essential revisions:

While the reviewers were in general positive about the research and the paper, a number of issues were raised by Reviewer #1 and Reviewer #2. Below are the main points, which are considered essential revisions and the authors should address point-by-point.

1) As seen in the tSNR map for macaques and marmosets, the tSNR around the primary visual cortex was much weaker than other cortices. Moreover, in marmosets, the EPI slices did not cover the entire brain and actually left most of the V1 uncovered. It would be better to analyze and discuss how the tSNR differences affect the present findings. For example, the author may consider including the tSNR as covariance in their map analysis.

2) Strengthen validity of the results: look into the EEG signals around the sensory cortex (e.g., V1) to see whether the findings in fMRI could be confirmed.

3) Clarifications:

a) As seen in Figure 2—figure supplement 2, there was a significant anticorrelation with burst-suppression at the ventricular borders. It is unclear whether the authors have done physiological or white matter/CSF/global nuisance regression as most of the rest-fMRI studies did.

b) Three different concentrations of the anesthetic sevoflurane were chosen for human participants. The authors found that the high concentration (3.9-4.6%) induced burst-suppression much better than the other two lower concentrations as expected. However, in rats, almost all asymmetric PCs were found at an intermediate concentration (2%) of isoflurane less at the low (1.5%) or high (2.5%) concentration in Rat 1. At the same time, all fMRI runs from Rat 2 with a 1.3% concentration of isoflurane had a prominent asymmetric PC. That is, it seems that only the high concentration of isoflurane could not induce burst-suppression well in rats, which was opposite to those findings in humans. Authors may explain what reasons may cause such differences and whether such differences may affect the major findings in differences between primates and rodents.

c) Authors found that some sensory areas in primates are excluded from those highly synchronized during the burst suppression. Clarify if each voxel in such areas shows burst suppression-like activity that is not synchronized with others. If this is the case, burst suppression can still be a global phenomenon. Because that in-ROI averaged time-series were used, it remains possible that each voxel inside the ROI is not synchronized but the ROI average shows burst suppression. Similarly, since that burst suppression should be defined by the existence of burst and suppressed periods, why not simply use this definition to check whether each voxel shows such an intermittent activity to evaluate whether it is a global phenomenon or not. We understand that this might be difficult to examine using non-human fMRI data, however, voxelwise analysis of human fMRI data should be conducted.

d) Why is there no synchronization during the slow-wave states under light anesthesia? During the slow-wave sleep, it is shown that the entire cortical network is decomposed into a modular-like network structure. Is there synchronization inside each module while no synchrony between modules?

---

## [Author Response]

Essential revisions:While the reviewers were in general positive about the research and the paper, a number of issues were raised by Reviewer #1 and Reviewer #2. Below are the main points, which are considered essential revisions and the authors should address point-by-point.1) As seen in the tSNR map for macaques and marmosets, the tSNR around the primary visual cortex was much weaker than other cortices. Moreover, in marmosets, the EPI slices did not cover the entire brain and actually left most of the V1 uncovered. It would be better to analyze and discuss how the tSNR differences affect the present findings. For example, the author may consider including the tSNR as covariance in their map analysis.

See response to Public Review Reviewer #1 major issue 1.

2) Strengthen validity of the results: look into the EEG signals around the sensory cortex (e.g., V1) to see whether the findings in fMRI could be confirmed.

See response to Public Review Reviewer #1 major issue 2.

3) Clarifications:a) As seen in Figure 2—figure supplement 2, there was a significant anticorrelation with burst-suppression at the ventricular borders. It is unclear whether the authors have done physiological or white matter/CSF/global nuisance regression as most of the rest-fMRI studies did.

See response to Public Review Reviewer #1 major issue 3.

b) Three different concentrations of the anesthetic sevoflurane were chosen for human participants. The authors found that the high concentration (3.9-4.6%) induced burst-suppression much better than the other two lower concentrations as expected. However, in rats, almost all asymmetric PCs were found at an intermediate concentration (2%) of isoflurane less at the low (1.5%) or high (2.5%) concentration in Rat 1. At the same time, all fMRI runs from Rat 2 with a 1.3% concentration of isoflurane had a prominent asymmetric PC. That is, it seems that only the high concentration of isoflurane could not induce burst-suppression well in rats, which was opposite to those findings in humans. Authors may explain what reasons may cause such differences and whether such differences may affect the major findings in differences between primates and rodents.

See response to Public Review Reviewer #1 major issue 4.

c) Authors found that some sensory areas in primates are excluded from those highly synchronized during the burst suppression. Clarify if each voxel in such areas shows burst suppression-like activity that is not synchronized with others. If this is the case, burst suppression can still be a global phenomenon. Because that in-ROI averaged time-series were used, it remains possible that each voxel inside the ROI is not synchronized but the ROI average shows burst suppression. Similarly, since that burst suppression should be defined by the existence of burst and suppressed periods, why not simply use this definition to check whether each voxel shows such an intermittent activity to evaluate whether it is a global phenomenon or not. We understand that this might be difficult to examine using non-human fMRI data, however, voxelwise analysis of human fMRI data should be conducted.

See responses to Public Review Reviewer #2 issues 1–2.

d) Why is there no synchronization during the slow-wave states under light anesthesia? During the slow-wave sleep, it is shown that the entire cortical network is decomposed into a modular-like network structure. Is there synchronization inside each module while no synchrony between modules?

See response to Public Review Reviewer #2 issue 3.